# Eluder dimension: localise it!

**Alireza Bakhtiari**
University of Alberta
sbakhtia@ualberta.ca

**Alex Ayoub**
University of Alberta
aayoub@ualberta.ca

**Samuel Robertson**
University of Alberta
smrobert@ualberta.ca

**David Janz**
University of Oxford
david.janz@stats.ox.ac.uk

**Csaba Szepesvári**
University of Alberta
szepesva@ualberta.ca

## Abstract

We establish a lower bound on the eluder dimension of generalised linear model classes, showing that standard eluder dimension-based analysis cannot lead to first-order regret bounds. To address this, we introduce a localisation method for the eluder dimension; our analysis immediately recovers and improves on classic results for Bernoulli bandits, and allows for the first genuine first-order bounds for finite-horizon reinforcement learning tasks with bounded cumulative returns.

## 1 Introduction

We study decision-making problems where the regret admits a first-order (small-cost) bound of the form

$$R_n \leq \sqrt{n\eta(a_\star)\Gamma_n} + \Gamma'_n,$$

with $\eta(a_\star)$ the optimal mean per-round cost and $\Gamma_n, \Gamma'_n$ instance and model dependent complexities.

The challenge in obtaining first-order bounds is in how we measure the complexity of the task. The (global) eluder dimension (Russo and Van Roy, 2013) is a standard complexity measure used to provide worst-case guarantees, but, as we shall argue, it ignores the local structure of the problem: analyses based on the global eluder dimension often introduce a factor in $\Gamma_n$ that scales like a per-step worst-case information/curvature parameter (denoted $\kappa$), which cancels out $\eta(a_\star)$ and destroys first-order gains. This subtle issue is present in much of the previous work on first-order bounds.

We show that by localising the eluder dimension—restricting it to a small neighbourhood of the optimal model—these $\kappa$ terms can be moved into the lower-order $\Gamma'_n$ term. This tightens the link between exploration and the actual difficulty of the instance and yields genuinely first-order bounds.

Our contributions may be summarised as follows:

1. *Localised $\ell_1$-eluder dimension.* We define a localised $\ell_1$-eluder dimension (in the sense of Liu et al., 2022) over a small-excess-loss neighbourhood, which avoids the $\kappa$ dependency in the classic generalised linear bandit setting (Filippi et al., 2010; Faury et al., 2020).

2. *Necessity of localisation.* We prove lower bounds showing that the global $\ell_1$- and $\ell_2$-eluder dimensions (Russo and Van Roy, 2013; Liu et al., 2022) must scale with $\kappa$ in the generalised linear setting, and that this negates small-cost or variance-dependent improvements.

3. *Stochastic bandits.* We propose a version-space optimistic algorithm, $\ell$-UCB, which takes a loss $\ell$ and builds confidence sets for the cost function. Under (a) bounded loss, (b) a

---

This is a corrected version of the originally published manuscript. The correction revises a definition and modifies the corresponding proofs accordingly. The main results and conclusions of the paper remain unchanged.

39th Conference on Neural Information Processing Systems (NeurIPS 2025).

Bernstein variance condition, and (c) a triangle condition (Foster and Krishnamurthy, 2021), $\ell$-UCB achieves a small-cost bound using analysis based on our localised eluder dimension.

4. *Reinforcement learning.* We extend our method to online RL, giving $\ell$-GOLF, and obtain the first $\kappa$-free first-order regret bound for finite-horizon RL with bounded rewards/costs.

## 1.1 Related work

*Small-cost in bandits.* Adversarial small-cost bounds are known (Neu, 2015; Allen-Zhu et al., 2018; Foster and Krishnamurthy, 2021; Ito et al., 2020; Olkhovskaya et al., 2023). However, adversarial algorithms are often conservative in stochastic regimes (Lattimore and Szepesvári, 2020, Ch. 18) and do not transfer cleanly to reinforcement learning, where optimism (Lai and Robbins, 1985) remains central for low regret (Ayoub et al., 2020; Weisz et al., 2023; Wu et al., 2025; Moulin et al., 2025).

In stochastic bandits, first-order bounds typically assume distributional knowledge (e.g., noise/cost models) (Abeille et al., 2021; Faury et al., 2022; Janz et al., 2024; Liu et al., 2024; Lee et al., 2024). We consider stochastic bandits with function approximation and *unknown* bounded cost distributions, aligning with adversarial-style uncertainty but in a stochastic environment.

*Small-cost in reinforcement learning.* Online first-order bounds have been shown by Wang et al. (2023) with the (strong) distributional Bellman completeness assumption. This assumption was then removed by Ayoub et al. (2024), but in the offline setting; Wang et al. (2024) extended this result back to the online setting. However, by relying on the global eluder dimension, Wang et al. (2023, 2024) suffer a (hidden) $\kappa$-dependence in the leading term that undermines first-order gains.

*Reward-first-order vs cost-first-order.* Reward-first-order bounds help when the optimal reward is small (Jin et al., 2020; Wagenmaker et al., 2022); this is very different from the cost-first-order guarantees we target (our results actually hold for both small-cost and small-reward settings). Small-cost results have been previously shown in structured settings such as tabular Markov decision processes (Lee et al., 2020) and linear-quadratic regulators (Kakade et al., 2020).

*Instance-optimal exploration.* Pure-exploration studies instance-dependent sample complexity, with notable works including policy-difference estimation for tabular reinforcement learning (Narang et al., 2024) and PEDEL for linear function approximation (Wagenmaker and Jamieson, 2022). The algorithm-agnostic lower bounds of Al-Marjani et al. (2022) show that PEDEL is near instance-optimal for tabular MDPs. These works are complementary to our regret-focused results.

## 2 Background on generalised linear models & loss functions

We collect the notation and standing assumptions for generalised linear models (GLMs) and losses used throughout. Fix a dimension $d \in \mathbf{N}_+$, and let $\mathcal{A}, \Theta \subset \mathbf{R}^d$ be closed sets, with $\Theta$ convex. Let $U \subset \mathbf{R}$ be a closed interval and let $\mu : U \to [0,1]$ be increasing. The GLM class with link $\mu$ and parameter set $\Theta$ is

$$\mathrm{GLM}(\mu, \Theta) = \{a \mapsto \mu(\langle a, \theta \rangle) \colon a \in \mathbf{R}^d, \theta \in \Theta, \langle a, \theta \rangle \in U\}.$$

We also consider losses $\ell \colon [0,1] \times [0,1] \to \mathbf{R}$, where $\ell(y, \hat{y})$ evaluates a prediction $\hat{y}$ against an outcome $y$. The next assumption records the structural conditions on $(\mathcal{A}, \Theta, \mu, \ell)$ used in our analysis; the final condition is satisfied when $\ell$ is the negative log-likelihood of the GLM associated with $\mu$.

**Assumption 1.** *We make the following assumptions:*

$$
\begin{array}{rll}
& \mathcal{A} \subset \mathbf{B}_2^d & \text{(action set bound)} \\
(\exists S > 0) & \Theta \subset S\mathbf{B}_2^d & \text{(parameter set bound)} \\
(\forall (a,\theta) \in \mathcal{A} \times \Theta) & \langle a, \theta \rangle \in U & \text{(valid domain)} \\
(\exists L > 0, \forall u, u' \in U) & |\mu(u) - \mu(u')| \leq L|u - u'| & (L\text{-Lipschitz link}) \\
(\exists M \geq 1, \forall u \in U^\circ) & |\ddot{\mu}(u)| \leq M\dot{\mu}(u) & (M\text{-self-concordant link}) \\
(\exists 1 \leq \kappa < \infty) & \kappa \geq \sup_{u \in U^\circ} 1/\dot{\mu}(u) & \text{(link derivative lower bound)} \\
(\forall y \in [0,1], \forall u \in U) & \partial_u \ell(y, \mu(u)) = \mu(u) - y. & \text{(link and loss are compatible)}
\end{array}
$$

**Remark 1.** *The requirement that $M, \kappa \geq 1$ in Assumption 1 is there solely to simplify our bounds.*

---

**Algorithm 1** the $\ell$-UCB bandit algorithm

---

**input** loss function $\ell$, model $\mathcal{F}$, nonnegative, nondecreasing confidence widths $(\beta_t)_{t \geq 1}$

**for** time-step $t \in \mathbf{N}_+$ **do**

    let $\mathcal{F}_t$ be the subset of the model given by

$$\mathcal{F}_t = \left\{ f \in \mathcal{F} \colon \sum_{i=1}^{t-1} \ell(Y_i, f(A_i)) \leq \inf_{\hat{f} \in \mathcal{F}} \sum_{i=1}^{t-1} \ell(Y_i, \hat{f}(A_i)) + \beta_t \right\},$$

    compute an optimistic function $f_t \in \mathcal{F}_t$ and action $A_t \in \mathcal{A}$ that satisfy

$$f_t(A_t) \leq f(a), \quad \forall (f, a) \in \mathcal{F}_t \times \mathcal{A},$$

    and play action $A_t$

**end for**

---

Examples of loss and link combinations that satisfy our assumptions include the log-loss with the sigmoid link function and the Poisson loss with the exponential link function:

**Example 1.** *The log-loss function $\ell_{\mathrm{X}}(y, p) = -y \log p - (1 - y) \log(1 - p)$ together with the sigmoid link function $\mu_{\mathrm{X}} \colon [-S, S] \to [0, 1]$ given by $u \mapsto 1/(1 + e^{-u})$ satisfies Assumption 1 with $L = 1/4$, $M = 1$ and $\kappa = 3e^S$.*

**Example 2.** *The Poisson loss function $\ell_{\mathrm{P}}(y, p) = p - y \log p$ together with the exponential link function $\mu_{\mathrm{P}} \colon [-S, 0] \to [0, 1]$ given by $u \mapsto e^u$ satisfies Assumption 1 with $L = M = 1$ and $\kappa = e^S$.*

## 3 Bandits with bounded costs and the $\ell$-UCB algorithm

Our bandit setting comprises a set of actions $\mathcal{A}$ and a corresponding set of action-dependent cost distributions $\mathcal{P} = \{P_a \colon a \in \mathcal{A}\}$ supported on the interval $[0, 1]$ (we will write $\mathrm{supp}\, P$ for the support of a measure $P$). At each round $t \in \mathbf{N}_+$, a learner selects an action $A_t \in \mathcal{A}$ and receives a cost $Y_t \sim P_{A_t}$. We measure the learner's performance over $n \in \mathbf{N}_+$ rounds by the $n$-step regret

$$R_n = \sum_{t=1}^{n} \eta(A_t) - \eta(a_\star) \quad \text{where} \quad \eta \colon a \mapsto \int y P_a(dy) \quad \text{and} \quad a_\star \in \operatorname*{arg\,min}_{a \in \mathcal{A}} \eta(a).$$

The learner may base its choice of $A_t$ on the past observations $A_1, Y_1, \ldots, A_{t-1}, Y_{t-1}$, any extra randomness independent of the observations (say, for tie-breaking), and prior knowledge in the form of a model class: a set $\mathcal{F}$ of functions $\mathcal{A} \to [0, 1]$ known to contain $\eta$. The key assumptions here are:

**Assumption 2** (Bounded costs)**.** *We have $\cup_{a \in \mathcal{A}} \mathrm{supp}\, P_a \subset [0, 1]$.*

**Assumption 3** (Realisability)**.** *We have that $\eta \in \mathcal{F}$.*

**Algorithm** Our algorithm, $\ell$-UCB (Algorithm 1) is an implementation of optimism with empirical risk minimisation-based confidence intervals.

At each time-step $t \in \mathbf{N}_+$, the algorithm constructs a confidence set $\mathcal{F}_t$ for $\eta$ composed of functions in $\mathcal{F}$ for which the empirical risk under the loss function $\ell \colon [0, 1] \times \mathcal{P} \to \mathbf{R}, \mathcal{P} \subset [0, 1]$ containing the image of the model, does not exceed that of the empirical risk minimiser by more than $\beta_t$, where $(\beta_t)_{t \geq 1}$ is a problem-dependent nonnegative, nondecreasing sequence of confidence widths. The algorithm then computes an optimistic function-action pair $(f_t, A_t) \in \mathcal{F}_t \times \mathcal{A}$ such that

$$f_t(A_t) \leq f(a), \quad \forall (f, a) \in \mathcal{F}_t \times \mathcal{A},$$

and plays $A_t$. Optimising over $\mathcal{F}_t \times \mathcal{A}$ is difficult without further assumptions. In Appendix A we detail a standard convex relaxation of this optimisation problem applicable to self-concordant models.

The crucial component to the $\ell$-UCB algorithm obtaining small-cost adaptivity is the right choice of the loss function $\ell$ used to construct the confidence intervals (and well-chosen confidence widths, based on the loss function and model class). Our requirements will be stated in the form of an assumption on the offset versions of the loss functions, and their expectations, which are defined thus:

**Definition 1.** Let $\mathcal{F}$ be a model class and $\ell\colon [0,1] \times [0,1] \to \mathbf{R}$ a loss function. For each $f \in \mathcal{F}$, we define the excess loss $\varphi_f\colon [0,1] \times \mathcal{A} \to \mathbf{R}$ and expected excess loss $\bar{\varphi}_f\colon \mathcal{A} \to \mathbf{R}_+$ as

$$\varphi_f(y,a) = \ell(y, f(a)) - \ell(y, \eta(a)) \quad \text{and} \quad \bar{\varphi}_f(a) = \int \varphi_f(\cdot, a) dP_a \,.$$

We will write $\Phi(\mathcal{F}) = \{\varphi_f\colon f \in \mathcal{F}\}$ and $\bar{\Phi}(\mathcal{F}) = \{\bar{\varphi}_f\colon f \in \mathcal{F}\}$ for the respective loss classes.

Let $\Delta\colon [0,1] \times [0,1] \to \mathbf{R}_+$ be the triangular discrimination function given by

$$\Delta(0,0) = 0 \quad \text{and} \quad \Delta(p,q) = \frac{(p-q)^2}{p+q} \quad \text{otherwise.}$$

**Assumption 4** (Loss function assumptions). *There exist constants $b, c, \gamma > 0$ such that for all $(f,a) \in \mathcal{F} \times \mathcal{A}$, letting $Y \sim P_a$, the following three bounds hold:*

$$|\varphi_f(Y,a)| \leq b \text{ a.s.}, \qquad \qquad \text{(bounded loss)}$$
$$\mathrm{Var}\,\varphi_f(Y,a) \leq c\bar{\varphi}_f(a), \qquad \qquad \text{(variance condition)}$$
$$\Delta(f(a), \eta(a)) \leq \gamma\bar{\varphi}_f(a). \qquad \qquad \text{(triangle condition)}$$

The first two conditions in Assumption 4, boundedness and the variance condition, allow for a Bernstein-type concentration on the excess loss class. The triangle condition is used in the regret decomposition to move from fast concentration to small-cost bounds. The conditions in Assumption 4 implicitly depend on $\eta$, and thus ought to hold uniformly for all $\eta \in \mathcal{F}$. Recall our two losses:

- Log-loss satisfies the triangle condition with $\gamma \leq 2$ (Proposition 19); for any $f \in \mathcal{F}$ such that $\|\varphi_f\|_\infty \leq b$, $\varphi_f$ satisfies the variance condition with $c = b + 4$ (Proposition 15).

- Poisson loss satisfies the triangle condition with $\gamma \leq 5$ (Proposition 20); for any $f \in \mathcal{F}$ such that $\|\varphi_f\|_\infty \leq b$, $\varphi_f$ satisfies the variance condition with $c = b + 2$ (Proposition 16).

The squared loss function fails to satisfy the triangle condition; Theorem 2 of Foster and Krishnamurthy (2021) shows that squared loss cannot lead to the small-cost bounds we seek.

## 4  The localised eluder dimension & first-order regret bounds for bandits

We now define the localised eluder dimension, which is a localisation of the (global) $\ell_1$-eluder dimension of Liu et al. (2022).

**Definition 2** (Localised eluder dimension). Let $\mathcal{Z}$ be a set, let $\Psi$ be a class of real-valued functions on $\mathcal{Z}$, and let $z = (z_1, z_2, \ldots, z_n)$ be a length-$n$ sequence in $\mathcal{Z}$. We define the following.

1. We say that $x \in \mathcal{Z}$ is $\varepsilon$-independent of $z$ with respect to $\Psi$ if there exists a $\psi \in \Psi$ such that

$$\sum_{t=1}^{n} |\psi(z_t)| \leq \varepsilon \qquad \text{and} \qquad |\psi(x)| > \varepsilon \,.$$

2. We say that $z$ is an $\varepsilon$-eluder sequence with respect to $\Psi$ if, for every $t \leq n$, the point $z_t$ is $\varepsilon$-independent of $z_1, \ldots, z_{t-1}$ with respect to $\Psi$.

3. The $(\varepsilon, \sigma)$-localised eluder dimension $\dim_{\mathrm{elud}}^{\sigma}(\varepsilon; \Psi)$ of $\Psi$ is the maximum length of an $\omega$-eluder sequence with respect to $\Psi$ over all $\omega \in [\varepsilon, \sigma]$.

**Remark 2.** *The localised eluder dimension $\dim_{\mathrm{elud}}^{\sigma}(\varepsilon; \Psi)$ is non-increasing in $\varepsilon$ and non-decreasing in $\sigma$. For each $\varepsilon \geq 0$, the global $\ell_1$-eluder dimension of Liu et al. (2022) at scale $\varepsilon$ is $\dim_{\mathrm{elud}}^{\infty}(\varepsilon; \Psi)$.*

**Remark 3.** *The distinction between $\ell_1$- and $\ell_2$-eluder dimensions is separate from localisation. Indeed, the $\ell_2$-eluder dimension of Russo and Van Roy (2013) at scale $\varepsilon$ of a function class $\Psi$ is equivalent to the global $\ell_1$-eluder dimension at scale $\varepsilon^2$ of the squared class $\Psi^2 := \{z \mapsto \psi(z)^2 \colon \psi \in \Psi\}$, namely $\dim_{\mathrm{elud}}^{\infty}(\varepsilon^2; \Psi^2)$. We think of the $\ell_1$-eluder dimension as being loss-agnostic, whereas the $\ell_2$-eluder dimension builds in the squared loss. The $\ell_1$ definition is thus more suitable for GLMs.*

The localised eluder dimension allows us to establish the following guarantee for Algorithm 1.

**Theorem 1** (Regret bound for $\ell$-UCB in bandits). *Fix $\delta \in (0,1)$, $n \in \mathbf{N}_+$, bandit instance $\mathcal{P}$, model class $\mathcal{F}$ and a loss function $\ell$. Suppose that $(\mathcal{P}, \mathcal{F}, \ell)$ satisfy Assumptions 2 to 4. Let $N_n$ denote the $1/n$-covering number of $\Phi(\mathcal{F})$ with respect to the uniform metric, and for each $t \in \mathbf{N}_+$ let*

$$\beta_t = 5/2 + 15(2b + c)\log(N_n h_t/\delta) \quad where \quad h_t = e + \log(1+t).$$

*Define*

$$d_n^\sigma = \dim_{\mathrm{elud}}^\sigma(1/n; \bar{\Phi}(\mathcal{F})) \quad and \quad \Gamma_n^\sigma = \gamma(1 + (d_n^\sigma + 1)b + 4d_n^\sigma \beta_n \log(1 + nb)).$$

*Suppose a learner uses Algorithm 1, $\ell$-UCB, over the course of $n$-many interactions with $\mathcal{P}$, with model class $\mathcal{F}$, loss function $\ell$ and confidence widths $(\beta_t)_{t \geq 1}$. Then, with probability at least $1 - \delta$,*

$$R_n \leq \inf_{\sigma \in [1/n, b]} \left\{ 3\sqrt{n\eta(a_\star)\Gamma_n^\sigma} + 6\Gamma_n^\sigma + \left(\frac{4\beta_n}{\sigma} + 1\right)\dim_{\mathrm{elud}}^\sigma(\sigma; \bar{\Phi}(\mathcal{F})) + 1 \right\}.$$

**Remark 4.** *The covering number $N_n$ featuring in the confidence widths $(\beta_t)_{t \geq 1}$ is independent of $\kappa$.*

The proof of Theorem 1 is located in Appendix D.

## 4.1 Why localisation matters: eluder dimension lower bound for generalised linear models

The following lower bound shows that any regret bound whose leading term scales with the global eluder dimension $\dim_{\mathrm{elud}}^\infty(\varepsilon; \bar{\Phi}(\mathcal{F}))$ necessarily incurs a worst-case curvature dependence. In contrast, localised quantities of the form $\dim_{\mathrm{elud}}^\sigma(\varepsilon; \bar{\Phi}(\mathcal{F}))$ with finite $\sigma$ can avoid this dependence. Thus, localisation is essential for obtaining first-order bounds in generalised linear settings.

**Theorem 2** (GLM $\ell_1$-eluder dimension lower bound). *Let $(\mu, \ell)$ satisfy the last four properties of Assumption 1 (link $L$-Lipschitz, $M$-self-concordant, link-derivative lower bound, and link–loss compatibility). Fix $S \geq 4/M$ and assume that $[-S, 0] \subset U$. Write*

$$\tilde{\kappa} = \frac{\dot{\mu}(0)}{2\dot{\mu}(-S/2)} \in (0, \infty), \qquad b = \min\{\lfloor S \rfloor, d - 1\}.$$

*Then, there exist $\mathcal{A} \subset \mathbf{B}_2^d$, $\Theta \subset S\mathbf{B}_2^d$, $\theta_\star \in \Theta$, and a bandit instance $\mathcal{P} = \{P_a : a \in \mathcal{A}\}$ such that $(\mathcal{A}, \Theta, \mu, \ell)$ satisfy Assumption 1, $\eta(a) = \mu(\langle a, \theta_\star \rangle)$, and $P_a = \delta_{\eta(a)}$ for all $a \in \mathcal{A}$. Writing $\mathcal{F} = \mathrm{GLM}(\mu, \Theta)$, for every $\varepsilon \leq \dot{\mu}(0)/(2M^2)$ the eluder dimension of the associated expected excess-loss class $\bar{\Phi}(\mathcal{F})$ satisfies*

$$\dim_{\mathrm{elud}}^\infty(\varepsilon; \bar{\Phi}(\mathcal{F})) \geq \frac{d-1}{4b}\exp\left\{\min\left(\frac{b}{16}, \frac{(\log(\tilde{\kappa}))_+^2}{8SM^2 + 4(\log(\tilde{\kappa}))_+}\right)\right\},$$

*for a sequence of actions taking values in $\mathcal{A}$, where $(x)_+ := \max\{x, 0\}$.*

The proof of Theorem 2 is given in Appendix G.

The quantity $\tilde{\kappa}$ compares the curvature at $0$ and at $-S/2$; for the sigmoid link it grows exponentially in $S$, and so the theorem yields an eluder lower bound that is already exponential in $S$.

**Corollary 3.** *Consider the setting of Theorem 2 with the log-loss $\ell_X$ and the sigmoid link function $\mu(u) = 1/(1 + e^{-u})$. Then, $M = 1$ and $\dot{\mu}(0) = 1/4$. Therefore, for $S \geq 4$, $d \geq 2$ and any $\varepsilon \leq 1/8$,*

$$\dim_{\mathrm{elud}}^\infty\left(\varepsilon; \bar{\Phi}(\mathrm{GLM}(\mu, \Theta))\right) \geq \frac{d-1}{4b}\exp\left\{\frac{\min\{\lfloor S \rfloor, d-1\}}{4300}\right\}.$$

The corollary follows by substituting the logistic quantities into Theorem 2; the proof is omitted.

To understand the implications of Theorem 2, consider the setting of logistic bandits in the usual low-information regime, where $\langle a_\star, \theta_\star \rangle \approx -S$; think clickthrough rates in online advertising, where even the best adverts rarely get clicked on. Then $\eta(a_\star) \approx \mu(\langle a_\star, \theta_\star \rangle) \approx \exp(-S)$, which suggests that our regret should be excellent; but at the same time the global eluder dimension can still be exponential in $S$, completely cancelling out the benefit of the $\eta(a_\star)$ small-cost term. This results in a bound that fails to truly adapt to the problem instance. We now show how localisation helps.

## 4.2 Regret upper bound with localisation for the generalised linear model setting

We now instantiate Theorem 1 for generalised linear models; see Appendix F.3 for the relevant proofs. The localised eluder dimension of $\bar{\Phi}(\mathrm{GLM}(\mu, \Theta))$ can be upper-bounded as follows:

**Proposition 4.** *Let Assumption 1 hold. Then, there exists a universal constant $C > 0$ such that*

$$\dim_{\mathrm{elud}}^{\sigma}(\varepsilon; \bar{\Phi}(\mathrm{GLM}(\mu, \Theta))) \leq Cd \log(1 + S^2 L/\varepsilon) \quad \text{for all} \quad 0 < \varepsilon \leq \sigma \leq 1/(4\kappa M^2).$$

Localisation thus allows for first-order bounds where the effect of the small-cost term $\eta(a_\star)$ is not overshadowed by $\kappa$. Proposition 4 yields the following specialisation of Theorem 1:

**Proposition 5** (Regret for $\ell$-UCB with the logistic model). *Let $\delta \in (0, 1)$, $S > 0$ and $n \in \mathbf{N}_+$. Consider the setting of Theorem 1, with the model class $\mathcal{F} = \mathrm{GLM}(\mu, \Theta)$ where $\mu(u) = 1/(1 + e^{-u})$ and the logistic loss function $\ell_{\mathrm{X}}$. Consider running $\ell$-UCB with confidence widths $(\beta_t)_{t \in \mathbf{N}_+}$ given by*

$$\beta_t = 5/2 + 60(3S + 1)\big[d \log(1 + 8Sn) + \log(h_t/\delta)\big], \qquad h_t = e + \log(1 + t).$$

*Then, for a constant $C > 0$, with probability at least $1 - \delta$, the resulting regret satisfies the bound*

$$R_n \leq C\sqrt{n\eta(a_\star)d\beta_n}\left(1 + \log(1 + Sn)\right) + Cd\beta_n\big[(1 + \log(1 + Sn))^2 + e^S S \log(1 + S)\big] + 12e^S.$$

The same result of Proposition 5 also holds, up to constant factors, for the Poisson model; each log-loss specific result used in the proof of Proposition 5 has a Poisson equivalent in Appendix C.

Observe that the regret bound of Proposition 5 holds as soon as Assumptions 1 to 3 are met. *Importantly, we do not assume that the observations are generated by a generalised linear model.* However, much of the literature does make that assumption, so we compare in that setting.

## 4.3 Discussion in the logistic bandit & maximum likelihood estimation settings

The maximum likelihood estimation (MLE) setting is the well-studied setting where the costs are sampled from a known generalised linear model (one might also call this a 'well-specified' setting).

A special case of the MLE setting with bounded rewards is the logistic bandit setting. Here, $\eta$ is given by a generalised linear model with the sigmoid link function and the responses are given by

$$Y_t \sim \mathrm{Bernoulli}(\eta(A_t)) \quad \text{for each } t \in \mathbf{N}_+.$$

In this setting, the leading term in the regret bound of Proposition 5 nearly matches the lower bound given by Abeille et al. (2021), which states that there exists a $C > 0$ such that

$$R_n \geq Cd\sqrt{nv(a_\star)} \quad \text{where} \quad v(a_\star) = \eta(a_\star)(1 - \eta(a_\star)).$$

Likewise, Proposition 5 almost matches the upper bounds of Faury et al. (2022) for their logistic-bandit-specific algorithm, which guarantee that for some $C > 0$, with probability at least $1 - \delta$,

$$R_n \leq CSd\sqrt{nv(a_\star)} \log(n/\delta) + CS^6 d\kappa(\log(n/\delta))^2.$$

The suboptimality of Proposition 5 here is in that it depends on $\eta(a_\star)$, providing only a small-cost bound, rather than on $v(a_\star)$; the latter allows for a simultaneous small-cost and small-reward bound. This is because Proposition 5 only assumes that the triangle condition is met on one side of the reward interval, and so only allows for small-cost bounds; strengthening the assumption to be two-sided (which is satisfied by the logistic model) would allow us to recover the $v(a_\star)$. (We do not do this, as it would rule out, for example, the Poisson GLM, which only gives small-cost bounds.)

Interestingly, while Faury et al. (2022) only consider the logistic bandit setting, their analysis actually shows a regret bound for the wider bounded reward setting. The distinction between our work and that of Faury et al. (2022) is that where we use an analysis-only localisation technique to move $\kappa$ to an additive term, Faury et al. (2022) use an explicit algorithmic warm-up procedure to do this. That is, they run an approximation of an optimal design at the start of interaction, until their confidence sets have shrunk to a neighbourhood of the true parameter (on the good event where the confidence sets do indeed contain the true parameter).[1] *The change from algorithmic localisation to analysis-only*

---

[1] Faury et al. (2022) also propose an online data-rejection procedure that can be used instead of a warm-up. This is, however, again, an algorithmic tool, in contrast to our analysis-only approach.

*localisation is vital for the upcoming reinforcement learning setting where, because we do not have random access to state-action pairs, the solving of an optimal design is not feasible.*

The works of Lee et al. (2024) and Emmenegger et al. (2024) also do away with the warm-up employed in Faury et al. (2022), using techniques based on likelihood ratios. However, they rely on their likelihood ratios forming a martingale, which restricts the results to the MLE setting.

**Remark 5** (Bernoullisation). *Any algorithm* A *that yields first-order regret for the logistic setting can be used to obtain first-order regret for the bounded reward setting using Bernoullisation. The trick is thus: for each time-step $t \in \mathbf{N}_+$, upon observing $Y_t \in [0, 1]$, we sample*

$$Y_t' \sim \text{Bernoulli}(Y_t)$$

*and feed $Y_t'$ to the algorithm* A. *Since the conditional means of $Y_t$ and $Y_t'$ are the same, first-order properties are preserved. Bernoullisation, however, destroys any second-order adaptivity of the algorithm. Indeed, consider the case where the $(Y_t)_{t \in \mathbf{N}_+}$ are equal to $1/2$ almost surely. Then, running empirical loss minimisation with the log-loss on $(Y_t)_{t \in \mathbf{N}_+}$ converges to $1/2$ after a single observation, but running the same procedure on the corresponding sequence $(Y_t')_{t \in \mathbf{N}_+}$ of independent* Bernoulli($1/2$) *random variables leads to an $\Omega(1/\sqrt{n})$ absolute error in the estimate.*

## 5 First-order regret bounds for online reinforcement learning

We consider the episodic reinforcement learning setting with horizon $H \in \mathbf{N}_+$. Let $M = (\mathcal{S}, \mathcal{A}, c, P, s_1)$ be a Markov decision process (MDP) with states $\mathcal{S}$, actions $\mathcal{A}$, a cost function $c = (c_1, \ldots, c_H)$ with $c_h \colon \mathcal{S} \times \mathcal{A} \to [0, 1]$, a deterministic starting state $s_1 \in \mathcal{S}$, and a transition kernel $P = (P_1, \ldots, P_H)$ with $P_h$ mapping from $\mathcal{S} \times \mathcal{A}$ to probability measures over $\mathcal{S}$.

The learner interacts with the MDP $M$ for $n \in \mathbf{N}_+$ episodes. At the start of each episode $t \in [n]$, the learner specifies a deterministic policy $\pi^t = (\pi_1^t, \ldots, \pi_H^t)$, where $\pi_h^t \colon \mathcal{S} \to \mathcal{A}$ for each $h \in [H]$. We allow the policy $\pi^t$ to depend on the states, actions and costs observed prior to the start of the $t$th episode, but not on the cost function $c$ or the dynamics $P$, as these are assumed to be unknown.

The learner's aim will be to minimise the expected cumulative cost incurred over the $n$ episodes. To formalise this, let $v_h^\pi$ be the value function of policy $\pi$ in $M$, given by

$$v_h^\pi(s) = \mathbf{E}_\pi \left[ \sum_{i=h}^H c_i(S_i, \pi_i(S_i)) \mid S_h = s \right],$$

for each $s \in \mathcal{S}$, where $\mathbf{E}_\pi[\,\cdot \mid S_h = s]$ denotes the expectation with respect to the states $S_h, \ldots, S_H$ induced by following the policy $\pi$ in the MDP $M$ starting at $S_h = s$. Then, letting $v_h^t := v_h^{\pi^t}$ ($h \in [H]$), the $n$-episode regret is given by

$$R_n = \sum_{t=1}^n v_1^t(s_1) - v_1^\star(s_1)$$

where $v_h^\star$ ($h \in [H]$) is the optimal value function, defined formally just after Eq. (1).

The key assumption that our learner will be allowed to exploit is the following:

**Assumption 5.** *Costs are nonnegative and sum to at most one over each episode.*

### 5.1 Preliminaries on Q-functions, Bellman optimality operators and greedy policies

Let $\mathcal{Q}$ be the set of all maps $\mathcal{S} \times \mathcal{A} \to [0, 1]^H$. For $q \in \mathcal{Q}$, we write $q_h$ for the map $(s, a) \mapsto [q(s, a)]_h$ (entry $h$ of $q(s, a)$), and we write $q^\wedge$ for the function $\mathcal{S} \to [0, 1]^H$ defined by

$$[q^\wedge(s)]_h = \min_{a \in \mathcal{A}} q_h(s, a) \quad \text{for all } s \in \mathcal{S} \text{ and } h \in [H].$$

We let $\mathcal{T} \colon \mathcal{Q} \to \mathcal{Q}$ denote the Bellman optimality operator for the MDP $M$, given stage-wise by

$$(\mathcal{T}q)_h(x) = c_h(x) + \int q_{h+1}^\wedge(s') P_h(ds' \mid x), \qquad x \in \mathcal{S} \times \mathcal{A}, \ h \in [H],$$

---
**Algorithm 2** The $\ell$-GOLF algorithm
---
**input** loss function $\ell$, models $\mathcal{F}$ and $\mathcal{G}$, nonnegative confidence widths $(\beta_t)_t$
**for** episode $t \in \mathbf{N}_+$ **do**
  for each $h \in [H]$ let

$$\mathcal{L}_h^{t-1}(u, g) = \sum_{i=1}^{t-1} \ell\big(1 \wedge \big(c_h(S_h^i, A_h^i) + u^\wedge(S_{h+1}^i)\big), g(S_h^i, A_h^i)\big)$$

  where $u^\wedge(s) = \min_{a \in \mathcal{A}} u(s, a)$, and let $\mathcal{F}^t$ be the subset of $\mathcal{F}$ given by

$$\mathcal{F}^t = \left\{ f \in \mathcal{F} : \mathcal{L}_h^{t-1}(f_{h+1}, f_h) \leq \inf_{g \in \mathcal{G}_h} \mathcal{L}_h^{t-1}(f_{h+1}, g) + \beta_t, \ \forall h \in [H] \right\},$$

  compute an optimistic function
$$f^t \in \arg\min_{f \in \mathcal{F}^t} f_1^\wedge(s_1)$$

  and play the policy $\pi^t := \pi^{f^t}$ greedy with respect to $f^t$
**end for**
---

with the convention $q_{H+1}^\wedge = 0$. We define the optimal action-value function $q^\star$ for $M$ to be the element of $\mathcal{Q}$ satisfying

$$\mathcal{T} q^\star = q^\star, \tag{1}$$

and define the value function $v^\star$ for $M$ to be $v^\star = q^{\star\wedge}$.

For any function $q \in \mathcal{Q}$, we write $\pi^q$ for the policy greedy with respect to $q$, defined by

$$\pi_h^q(s) \in \arg\min_{a \in \mathcal{A}} q_h(s, a) \quad \text{for all } s \in \mathcal{S} \text{ and } h \in [H].$$

## 5.2 The $\ell$-GOLF algorithm, model and loss assumptions & regret bound

Our $\ell$-GOLF algorithm (Algorithm 2) is an extension of $\ell$-UCB to the episodic online reinforcement learning setting, generalising the GOLF algorithm of Jin et al. (2021) to arbitrary loss functions (GOLF is recovered by taking $\ell$ to be the squared loss). The algorithm requires the specification of a loss function $\ell \colon [0, 1]^2 \to \mathbf{R}$, confidence widths $(\beta_t)_{t \in [n]}$ and function classes $\mathcal{F}, \mathcal{G} \subset \mathcal{Q}$. The model $\mathcal{F}$ contains candidate action-value functions for estimating $q^\star$, while the model $\mathcal{G}$ contains candidate Bellman updates; we write $\mathcal{G}_h = \{g_h \colon g \in \mathcal{G}\}$ for the stage-$h$ slice of $\mathcal{G}$. For convenience, we augment every $f \in \mathcal{F} \cup \mathcal{G}$ with $f_{H+1} = 0$.

Now, in each episode $t \in \mathbf{N}_+$, the algorithm constructs a confidence set $\mathcal{F}^t \subset \mathcal{F}$ containing action-value functions that are close to satisfying the Bellman optimality condition $f = \mathcal{T}f$ on the data observed thus far, with errors penalised according to $\ell$. It then selects an optimistic function $f^t \in \mathcal{F}^t$, and plays the policy $\pi^t := \pi^{f^t}$ greedy with respect to $f^t$.

We make the following realisability and generalised completeness assumptions of Antos et al. (2008):

**Assumption 6** (Realisability). *We assume that $q^\star \in \mathcal{F}$.*

**Assumption 7** (Generalised completeness). *We assume that $\mathcal{T}\mathcal{F} \subset \mathcal{G}$.*

**Definition 3.** For any $f \in \mathcal{F}$, $h \in [H]$, $x \in \mathcal{S} \times \mathcal{A}$ and $s' \in \mathcal{S}$, we let

$$y_h^f(x, s') = 1 \wedge (c_h(x) + f_{h+1}^\wedge(s'))$$

be the response under the model $f$. With the same symbols, we define the excess Bellman loss function

$$\varphi_h^f(x, s') = \ell(y_h^f(x, s'), f_h(x)) - \ell(y_h^f(x, s'), (\mathcal{T}f)_h(x)),$$

and the expected excess Bellman loss function

$$\bar{\varphi}_h^f(x) = \int \varphi_h^f(x, s') \, P_h(ds' \mid x).$$

With that, our assumptions on the loss function for the reinforcement learning setting mirror the bandit assumptions:

**Assumption 8** (RL loss function assumptions). *There exist constants $b, c, \gamma > 0$ such that for all $f \in \mathcal{F}$, $h \in [H]$, $x \in \mathcal{S} \times \mathcal{A}$, $S' \sim P_h(x)$, the following hold:*

$$|\varphi_h^f(x, S')| \leq b \text{ a.s.}, \qquad\qquad\qquad \text{(RL boundedness)}$$

$$\mathrm{Var}\, \varphi_h^f(x, S') \leq c\bar{\varphi}_h^f(x), \qquad\qquad\qquad \text{(RL variance condition)}$$

$$\Delta(f_h(x), (\mathcal{T}f)_h(x)) \leq \gamma\bar{\varphi}_h^f(x). \qquad\qquad\qquad \text{(RL triangle condition)}$$

**Theorem 6.** *Fix $\delta \in (0, 1)$, $n \in \mathbf{N}_+$, MDP $M$, model classes $\mathcal{F}$ and $\mathcal{G}$ and a loss function $\ell$. Suppose that $(M, \mathcal{F}, \mathcal{G}, \ell)$ satisfy Assumptions 6 to 8. Let*

$$\Phi_{\mathrm{RL}}(\mathcal{F}) = \{\varphi_h^f \colon h \in [H],\ f \in \mathcal{F}\},$$

*let $N_n$ be the $1/n$-covering number of $\Phi_{\mathrm{RL}}(\mathcal{F})$ with respect to the uniform metric, let $h_t = e + \log(1 + t)$ for each $t \in [n]$, and let*

$$\beta_t = 5/2 + 15(3b + c)\log(HN_nh_t/\delta), \qquad t \in \mathbf{N}_+.$$

*For each $f \in \mathcal{F}$ and $h \in [H]$, let $\mu_h^f$ denote the stage-$h$ state-action occupancy measure induced by following the policy $\pi^f$, and define*

$$\Psi_h(\mathcal{F}) = \left\{\nu \mapsto \int \bar{\varphi}_h^g \, d\nu \colon g \in \mathcal{F}\right\},$$

*viewed as a class of $[0, b]$-valued functions on $\{\mu_h^f \colon f \in \mathcal{F}\}$. For $\sigma \in [1/n, b]$, define*

$$d_n^\sigma = \sum_{h=1}^H \mathrm{dim}_{\mathrm{elud}}^\sigma(1/n; \Psi_h(\mathcal{F})) \quad and \quad \tilde{d}_n^\sigma = \sum_{h=1}^H \mathrm{dim}_{\mathrm{elud}}^\sigma(\sigma; \Psi_h(\mathcal{F})),$$

*and*

$$\Gamma_n^\sigma = \gamma\big[H + (d_n^\sigma + H)b + 4d_n^\sigma\beta_n\log(1 + nb)\big].$$

*Suppose a learner uses Algorithm 2, $\ell$-GOLF, over the course of $n$-many episodes with $M$, model classes $\mathcal{F}$ and $\mathcal{G}$, loss function $\ell$ and confidence widths $(\beta_t)_{t \in \mathbf{N}_+}$. Then, with probability at least $1 - \delta$,*

$$R_n \leq \inf_{\sigma \in [1/n, b]} \left\{5\sqrt{Hnv_1^\star(s_1)\Gamma_n^\sigma} + 12H\Gamma_n^\sigma + \left(\frac{4\beta_n}{\sigma} + 1\right)\tilde{d}_n^\sigma + H\right\}.$$

Theorem 6 is established in Appendix E. It is the RL analogue of Theorem 1, with the bandit class $\bar{\Phi}(\mathcal{F})$ replaced by the stage-wise occupancy-measure classes $\Psi_h(\mathcal{F})$. For context, the closest results to ours are those of Wang et al. (2023, 2024) for online RL. Both provide a small-cost regret bound scaling with the Bellman eluder dimension; however, without our notion of a *localised* dimension, their regret bound scales with $\kappa$ in the leading term for logistic linear models. This entirely offsets any benefit of their small-cost analysis; the bound is not truly instance-adaptive. Moreover, Wang et al. (2023) assumes that the distributional Bellman operator (Bellemare et al., 2017) lies in their model class, an assumption that is significantly stronger than our Assumption 7 (as discussed in Ayoub et al., 2024). An argument for extending the results from costs to rewards was given in Ayoub et al. (2025).

## 6 Conclusion

We have shown that standard eluder dimension analysis inherently fails to achieve first-order regret bounds in generalised linear model settings. By introducing the localised $\ell_1$-eluder dimension, we overcome this limitation, removing problematic worst-case dependencies and achieving genuinely adaptive, first-order regret bounds. Our refined analysis recovers and sharpens classical results in Bernoulli bandit scenarios and demonstrates clear practical advantages through the $\ell$-UCB algorithm.

Moreover, our localisation approach successfully extends to finite-horizon reinforcement learning via the $\ell$-GOLF algorithm, providing the first genuine first-order regret bounds in this setting. This highlights the crucial role of localisation techniques in developing instance-adaptive algorithms, opening promising avenues for further exploration in broader learning contexts.

## Acknowledgements

The authors thank Marc Abeille for pointing out a serious error in an earlier version of this manuscript. Alex Ayoub gratefully acknowledges funding from Netflix. Csaba Szepesvári gratefully acknowledges funding from the Canada CIFAR AI Chairs Program, Amii, NSERC and Netflix.

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

# Appendices

# A Self-concordance & convex relaxation

Take a parametric model class $\mathcal{F} = \{f_\theta \colon \theta \in \Theta\}$ where $\Theta \subset \mathbf{R}^d$ is a convex parameter set satisfying $\|\theta\|_2 \leq S$ for all $\theta \in \Theta$, for some $S > 0$. Let $\mathcal{Y} \subset \mathbf{R}$. For any $(y, a) \in \mathcal{Y} \times \mathcal{A}$, let $\ell_{(y,a)} \colon \mathbf{R}^d \to \mathbf{R}$ be given by $\theta \mapsto \ell(y, f_\theta(a))$. Consider the following self-concordance assumption.

**Assumption 9** (Self-concordance of losses). *Assume that for all $z \in \mathcal{Y} \times \mathcal{A}$, $\ell_z$ is convex and thrice differentiable. Moreover, assume that there exists an $M > 0$ such that for all $z \in \mathcal{Y} \times \mathcal{A}$, $\theta \in \Theta^\circ$ (the interior of the convex set $\Theta$) and $u, v \in \mathbf{R}^d$,*

$$|\langle D_u^3 \ell_z(\theta) v, v\rangle| \leq M \|u\|_2 \langle \nabla^2 \ell_z(\theta) v, v\rangle \,,$$

*where $D_u^3 \ell_z(\theta) \in \mathbf{R}^{d \times d}$ denotes the third directional derivative of $\ell_z$ at in the direction $u$ evaluated at $\theta$, and $\nabla^2 \ell_z(\theta) \in \mathbf{R}^{d \times d}$ is a matrix of the second order partial derivatives of $\ell_z$ evaluated at $\theta$.*

In particular, the generalised linear models introduced in Example 1 and Example 2 are $M = 1$ self-concordant (Faury et al., 2020; Lee et al., 2024). As shown in Janz et al. (2024), Assumption 9 is equivalent to requiring that $|\ddot{\mu}(x)| \leq M \dot{\mu}(x)$ for all $x$ in the domain of $\mu$, which holds for these GLMs. Moreover, a recent result by Liu et al. (2024) shows that many GLMs satisfy Assumption 9.

Let $\mathcal{L}_t(\theta) = \sum_{i=1}^{t-1} \ell(Y_i, f_\theta(A_i))$ be the empirical risk for a parameter $\theta \in \Theta$ on the first $t - 1$ observations, and $\hat{\theta}_t \in \Theta$ be an ERM. Consider the confidence sets of the form

$$\Theta_t = \{\theta \in \Theta \colon \mathcal{L}_t(\theta) - \mathcal{L}_t(\hat{\theta}_t) \leq \beta_t\} \,,\ \beta_t > 0 \,,\ t \in \mathbf{N}_+ \,.$$

These can be enclosed within an ellipsoid as follows.

**Theorem 7.** *Under Assumption 9, for all $t \in \mathbf{N}_+$,*

$$\Theta_t \subset \{\theta \in \Theta \colon \|\theta - \hat{\theta}_t\|^2_{\nabla^2 \mathcal{L}_t(\hat{\theta}_t)} \leq 2(1 + SM)\beta_t\} \,.$$

We provide a proof for completeness, but this result is well known (see, for example, Lee et al., 2024).

**Lemma 8** (Proposition 10 of Sun and Tran-Dinh (2019)). *Let $g(x) = \frac{\exp(x) - x - 1}{x^2}$. For any $\theta, \theta' \in \Theta$, under Assumption 9,*

$$g(-M\|\theta - \theta'\|_2) \|\theta - \theta'\|^2_{\nabla^2 \mathcal{L}_t(\theta')} \leq \mathcal{L}_t(\theta) - \mathcal{L}_t(\theta') - \langle \nabla \mathcal{L}_t(\theta'), \theta - \theta'\rangle$$
$$\leq g(M\|\theta - \theta'\|_2) \|\theta - \theta'\|^2_{\nabla^2 \mathcal{L}_t(\theta')} \,.$$

*Proof of Theorem 7.* From Lemma 8, and observing that since $\hat{\theta}_t$ is an ERM and $\Theta$ is convex, $\langle \nabla \mathcal{L}_t(\hat{\theta}_t), \theta - \hat{\theta}_t\rangle$ is nonnegative for any $\theta \in \Theta$, we have that for any $\theta \in \Theta$,

$$g(-M\|\theta - \hat{\theta}_t\|_2) \|\theta - \hat{\theta}_t\|^2_{\nabla^2 \mathcal{L}_t(\hat{\theta}_t)} \leq \mathcal{L}_t(\theta) - \mathcal{L}_t(\hat{\theta}_t) \,.$$

Using that $\frac{\exp(x) - x - 1}{x^2} \geq \frac{1}{-x + 2}$ whenever $x \leq 0$, bounding $\|\theta - \hat{\theta}_t\|_2 \leq 2S$ and staring at the result a little ought to convince the reader of the veracity of the claim. ∎

# B  A uniform Bernstein concentration inequality

The following uniform Bernstein inequality, proven over the course of this section, will be needed to prove both the bandit and the reinforcement learning regret bounds. It is the use of this inequality that necessitates the variance condition and boundedness condition for the excess loss classes.

**Theorem 9** (Uniform Bernstein inequality). *Let $\mathcal{Z}$ be a set, $(Z_t)_t$ be a $\mathcal{Z}$-valued process adapted to a filtration $(\mathbf{F}_t)_t$, and $\Phi$ a set of real-valued functions on $\mathcal{Z}$. Assume that:*

1. *For some $b > 0$, for all $\varphi \in \Phi$, $t \in \mathbf{N}_+$, $|\mathbf{E}[\varphi(Z_t) \mid \mathbf{F}_{t-1}] - \varphi(Z_t)| \leq b$.*

2. *For some $c > 0$, for all $\varphi \in \Phi$ and $t \in \mathbf{N}_+$, $\mathrm{Var}(\varphi(Z_t) \mid \mathbf{F}_{t-1}) \leq c\mathbf{E}[\varphi(Z_t) \mid \mathbf{F}_{t-1}]$.*

*Let $\delta \in (0,1)$, $\varepsilon > 0$ and let $N$ be the $\varepsilon$-covering number of $\Phi$ in the uniform metric. For any $n \in \mathbf{N}_+$, define*

$$\beta(n, \delta, \varepsilon, N) = \frac{5n\varepsilon}{2} + 15(b+c)\log(Nh_n/\delta),$$

*where $h_n = e + \log(1+n)$. Then, with probability at least $1 - \delta$, for all $\varphi \in \Phi$ and $n \in \mathbf{N}_+$,*

$$\sum_{t=1}^{n} \mathbf{E}[\varphi(Z_t) \mid \mathbf{F}_{t-1}] \leq 2\sum_{t=1}^{n} \varphi(Z_t) + 2\beta(n, \delta, \varepsilon, N).$$

To prove Theorem 9, we will need the following definitions and results:

**Definition 4** (CGF-like). We say a twice differentiable function $\psi : [0, \lambda_{\max}) \to \mathbf{R}_+$ is *CGF-like* if $\psi$ is strictly convex, $\psi(0) = \psi'(0) = 0$ and $\psi''(0)$ exists.

**Definition 5** (sub-$\psi$ process). Let $\mathbf{F}$ be a filtration, $\psi : [0, \lambda_{\max}) \to \mathbf{R}_+$ be a CGF-like function and let $(S_t)_t$ and $(V_t)_t$ be respectively $\mathbf{R}$-valued and $\mathbf{R}_+$-valued $\mathbf{F}$-adapted processes. We say that $(S_t, V_t)_t$ is a sub-$\psi$ process if, for every $\lambda \in [0, \lambda_{\max})$, there exists an $\mathbf{F}$-adapted supermartingale $L(\lambda)$ such that

$$M_t(\lambda) := \exp\{\lambda S_t - \psi(\lambda)V_t\} \leq L_t(\lambda) \quad \text{almost surely for all } t \geq 0.$$

**Definition 6** (Sub-gamma process). We say that a random process $(S_t, V_t)_t$ is sub-gamma with parameter $\vartheta > 0$ if it is sub-$\psi$ for the CGF-like function $\psi \colon [0, 1/\vartheta) \to \mathbf{R}_+$ mapping $\lambda \mapsto \frac{\lambda^2}{2(1-\vartheta\lambda)}$.

**Theorem 10** (Sub-gamma concentration). *For a sub-gamma process $(S_t, V_t)_t$ with parameter $c > 0$, and any $\rho > 0$ and $\delta \in (0,1)$, with probability at least $1 - \delta$, for all $t \geq 1$,*

$$S_t \leq 4\sqrt{V_t \log(H_t/\delta)} + 11(c+\rho)\log(H_t/\delta) \quad \text{where} \quad H_t = \log(1 + V_t/\rho^2) + e.$$

Theorem 10 is a consequence of Theorem 3.1 of Whitehouse et al. (2023); we will prove it shortly.

**Proposition 11.** *Let $\mathbf{F}$ be a filtration and let $(X_t)_{t \in \mathbf{N}_+}$ be a square-integrable, $\mathbf{F}$-adapted process satisfying*

$$X_t \leq \mathbf{E}[X_t \mid \mathbf{F}_{t-1}] + b \quad \text{a.s. for all } t \in \mathbf{N}_+.$$

*Then, for*

$$S_t = \sum_{i=1}^{t} X_i - \mathbf{E}[X_i \mid \mathbf{F}_{i-1}] \quad \text{and} \quad V_t = \sum_{i=1}^{t} \mathrm{Var}(X_i \mid \mathbf{F}_{i-1}),$$

*the process $(S_t, V_t)_{t \in \mathbf{N}_+}$ is sub-gamma with parameter $b/3$.*

*Proof of Proposition 11.* If the random variables $(X_t - \mathbf{E}[X_t \mid \mathcal{F}_{t-1}])_{t \in \mathbf{N}_+}$ are independent, the result follows directly from Theorem 2.10 (Bernstein's inequality) in Boucheron et al. (2013) combined with the discussion immediately after Corollary 2.11 therein. For the adapted result, use the tower property of the conditional expectation to extend the independent case. ∎

*Proof of Theorem 9.* We will write $\mathbf{E}_{t-1}$ and $\mathrm{Var}_{t-1}$ to denote $\mathbf{F}_{t-1}$-conditional expectation and variance operators, respectively. Now let $\Phi(\varepsilon) \subset \Phi$ be a uniform $\varepsilon$-cover of $\Phi$ with cardinality $N$. Then for any $\varphi \in \Phi$ there exists some $\hat{\varphi} \in \Phi(\varepsilon)$, such that for any $n \in \mathbf{N}_+$,

$$\sum_{t=1}^{n} \mathbf{E}_{t-1}\varphi(Z_t) - \varphi(Z_t) \leq 2n\varepsilon + \sum_{t=1}^{n} \mathbf{E}_{t-1}\hat{\varphi}(Z_t) - \hat{\varphi}(Z_t).$$

Now observe that for any $\hat{\varphi} \in \Phi(\varepsilon)$, applying Proposition 11 to the process $X_t = -\hat{\varphi}(Z_t)$ and using our assumptions on $\Phi$, we have that

$$\left( \sum_{i=1}^{t} \mathbf{E}_{i-1} \hat{\varphi}(Z_i) - \hat{\varphi}(Z_i), \sum_{i=1}^{t} \mathrm{Var}_{i-1} \hat{\varphi}(Z_i) \right)_{t \in \mathbf{N}_+}$$

is a sub-gamma process with parameter $b/3$. Applying Theorem 10 with $\rho = b$ and a confidence parameter $\delta/N$, and taking a union bound over the $N$ functions in $\Phi(\varepsilon)$, we conclude that with probability at least $1 - \delta$,

$$\sum_{t=1}^{n} \mathbf{E}_{t-1} \hat{\varphi}(Z_t) - \hat{\varphi}(Z_t) \leq 4 \sqrt{\sum_{i=1}^{n} \mathrm{Var}_{i-1} \hat{\varphi}(Z_i) \log(Nh_n/\delta)} + \frac{44b}{3} \log(Nh_n/\delta)$$

where we have upper bounded the $H_n$ therein, defined as in Theorem 10, by $h_n = e + \log(1 + n)$. Next, by the variance condition and Young's inequality,

$$4 \sqrt{\sum_{i=1}^{n} \mathrm{Var}_{i-1} \hat{\varphi}(Z_i) \log(Nh_n/\delta)} \leq \frac{1}{2} \sum_{t=1}^{n} \mathbf{E}_{t-1} \hat{\varphi}(Z_t) + 8c \log(Nh_n/\delta) \,.$$

We arrive at the desired result by bounding $\mathbf{E}_{t-1} \hat{\varphi}(Z_t) \leq \mathbf{E}_{t-1} \varphi(Z_t) + \varepsilon$, combining this with the previous inequalities and bounding the constants slightly for convenience. ∎

We now prove Theorem 10, which is an application of the following result:

**Theorem 12** (Theorem 3.1, Whitehouse et al. (2023)). *Let $(S_t, V_t)_{t \geq 0}$ be a sub-$\psi$ process for a CGF-like function $\psi : [0, c) \to \mathbf{R}_+$ satisfying $\lim_{\lambda \uparrow c} \psi'(\lambda) = \infty$. Let $\alpha > 1$, $\beta > 0$, $\delta \in (0, 1)$ and let $h : \mathbf{R}_+ \to \mathbf{R}_+$ be an increasing function such that $\sum_{k \in \mathbf{N}} 1/h(k) \leq 1$. Define the function $\ell_\beta : \mathbf{R}_+ \to \mathbf{R}_+$ by*

$$\ell_\beta(v) = \log h \left( \log_\alpha \left( \frac{v \vee \beta}{\beta} \right) \right) + \log \left( \frac{1}{\delta} \right),$$

*where, for brevity, we have suppressed the dependence of $\ell_\beta$ on $(\alpha, \delta, h)$. Then*

$$\mathbf{P} \left( \exists t \geq 0 \colon S_t \geq (V_t \vee \beta) \cdot (\psi^*)^{-1} \left( \frac{\alpha}{V_t \vee \beta} \ell_\beta(V_t) \right) \right) \leq \delta \,,$$

*where $\psi^*$ is the convex conjugate of $\psi$.*

*Proof of Theorem 10.* The result follows from applying Theorem 12 to our sub-gamma process with $\alpha = e$, $\beta = \rho^2$ and $h(k) = (k + e)^2$, and bounding the result crudely. In particular, for our choices of $\alpha$ and $h$, we have the bound

$$\ell_{\rho^2}(V_t) = \log(\log(\rho^{-2} V_t \vee 1) + e)^2 + \log 1/\delta \leq 2 \log((\log(1 + V_t/\rho^2) + e)/\delta) = 2 \log(H_t/\delta) \,.$$

Now, since for our choice of $\psi$, $\psi^{*-1}(t) = \sqrt{2t} + tc$, the bound from Theorem 12 can be further bounded as

$$\begin{aligned}
(V_t \vee \beta) \cdot (\psi^*)^{-1} \left( \frac{\alpha}{V_t \vee \beta} \ell_\beta(V_t) \right) &= \sqrt{2e(V_t \vee \rho^2) \ell_{\rho^2}(V_t)} + ec \ell_{\rho^2}(V_t) \\
&\leq 2 \sqrt{e(V_t \vee \rho^2) \log(H_t/\delta)} + 2ec \log(H_t/\delta) \\
&\leq 2 \sqrt{eV_t \log(H_t/\delta)} + 2(\rho\sqrt{e} + ce) \log(H_t/\delta) \\
&\leq 4 \sqrt{V_t \log(H_t/\delta)} + 11(c + \rho) \log(H_t/\delta) \,,
\end{aligned}$$

where the penultimate inequality uses that for $a, b > 0$, $\sqrt{a \vee b} \leq \sqrt{a + b} \leq \sqrt{a} + \sqrt{b}$ and that since $\log(H_t/\delta) \geq 1$, $\sqrt{\log(H_t/\delta)} \leq \log(H_t/\delta)$. ∎

# C    Analysis of the log-loss and Poisson loss functions

For convenience, we restate our loss function conditions:

**Assumption 4** (Loss function assumptions). *There exist constants $b, c, \gamma > 0$ such that for all $(f, a) \in \mathcal{F} \times \mathcal{A}$, letting $Y \sim P_a$, the following three bounds hold:*

$$|\varphi_f(Y, a)| \leq b \text{ a.s.}, \qquad\qquad\qquad \text{(bounded loss)}$$
$$\mathrm{Var}\, \varphi_f(Y, a) \leq c\bar{\varphi}_f(a), \qquad\qquad\qquad \text{(variance condition)}$$
$$\Delta(f(a), \eta(a)) \leq \gamma\bar{\varphi}_f(a). \qquad\qquad\qquad \text{(triangle condition)}$$

We now establish that the variance condition and triangle condition hold for the log-loss excess loss class $\Phi_X$ induced by the loss function $\ell_X$ and the Poisson loss excess loss class $\Phi_P$ induced by $\ell_P$.

## C.1    Establishing the variance condition

For our proof of the variance condition, we will assume that all $\varphi \in \Phi_X \cup \Phi_P$ satisfy the pointwise bound

$$\|\varphi\|_\infty \leq b.$$

This being satisfied relies on the choice of the model class $\mathcal{F}$. In Appendix F.1, we will verify that for a compatible GLM with parameter norm $S > 0$, the boundedness condition holds with $b = 4S$.

To establish the variance condition, we will use the following result of Erven et al. (2012), and in particular a special case stated and proven immediately afterwards.

**Lemma 13** (Lemma 10, Erven et al. (2012)). *Let $g(x) = (e^x - x - 1)/x^2$ for $x \neq 0$ and $g(0) = 1/2$, and let $X$ be a random variable satisfying $|X| \leq b$. Then, for all $t > 0$, there exists a $C_t \geq g(-tb)$ such that*

$$\mathbf{E}X = \frac{1}{t}(1 - \mathbf{E}e^{-tX}) + C_t t \mathbf{E}X^2.$$

**Lemma 14.** *Suppose that $X$ is a random variable satisfying*

1. *Boundedness: $|X| \leq b < \infty$; and*

2. *Stochastic mixability: $\mathbf{E}[\exp\{-X/2\}] \leq 1$.*

*Then, $\mathrm{Var}\, X \leq (b + 4)\mathbf{E}X$.*

*Proof of Lemma 14.* Applying Lemma 13, with $t = 1/2$, we obtain that there exists a $C \geq g(-b/2)$ such that

$$\mathbf{E}X \geq 2(1 - \mathbf{E}e^{-X/2}) + \frac{C}{2}\mathbf{E}X^2 \geq \frac{C}{2}\mathbf{E}X^2 \geq \frac{g(-b/2)}{2}\mathbf{E}X^2.$$

The result follows by using the numerical inequality $g(x) \geq 1/(2 - x)$ that holds for all $x \leq 0$; that $\mathbf{E}X^2 \geq \mathrm{Var}\, X$ for every random variable with a finite variance; and rearranging. ∎

We are now ready to prove the variance condition for the log-loss and Poisson loss functions.

**Proposition 15** (Log-loss variance condition). *Every $\varphi \in \Phi_X$ uniformly bounded by $b > 0$ satisfies the variance condition with constant $c = b + 4$.*

*Proof.* The result follows from Lemma 14 combined with that every $\varphi \in \Phi_X$ is stochastically mixable, which we establish now. Fix $\varphi \in \Phi_X$ and observe that $\varphi$ is induced by some $f \in \mathcal{F}$. Now fix $a \in \mathcal{A}$, let

$$p = f(a), \qquad q = \eta(a),$$

and let $Y \sim P_a$, so that $\mathbf{E}[Y] = q$.

If $q \in \{0, 1\}$, then $Y = q$ almost surely, and hence

$$\mathbf{E}\exp\{-\varphi(Y, a)/2\} = \begin{cases} \sqrt{1 - p}, & q = 0, \\ \sqrt{p}, & q = 1, \end{cases}$$

which is at most 1. It remains to consider the case $p, q \in (0, 1)$.

In this case, for all $y \in [0, 1]$,

$$\varphi(y, a) = -\log\left(\frac{p}{q}\right)^y - \log\left(\frac{1-p}{1-q}\right)^{1-y},$$

and therefore

$$
\begin{aligned}
\mathbf{E}\exp\{-\varphi(Y, a)/2\} &= \mathbf{E}\left[\left(\sqrt{\frac{p}{q}}\right)^Y \left(\sqrt{\frac{1-p}{1-q}}\right)^{1-Y}\right] \\
&\leq \mathbf{E}\left[Y\sqrt{\frac{p}{q}} + (1-Y)\sqrt{\frac{1-p}{1-q}}\right] \quad \text{(weighted AM-GM, since } Y \in [0, 1]) \\
&= q\sqrt{\frac{p}{q}} + (1-q)\sqrt{\frac{1-p}{1-q}} \\
&= \sqrt{pq} + \sqrt{(1-p)(1-q)} \\
&\leq 1, \quad\quad\quad\quad\quad\quad\quad\quad\quad\quad\quad\quad\quad\quad\quad\quad\quad\quad \text{(Cauchy-Schwarz)}
\end{aligned}
$$

and so $\varphi$ is stochastically mixable.

Hence, by Lemma 14,

$$\operatorname{Var}\varphi(Y, a) \leq (b+4)\mathbf{E}[\varphi(Y, a)] = (b+4)\bar{\varphi}(a),$$

which is the desired variance condition. ∎

**Proposition 16** (Poisson loss variance condition). *Every $\varphi \in \Phi_{\mathrm{P}}$ uniformly bounded by $b > 0$ satisfies the variance condition with $c = b + 2$.*

*Proof.* We establish that for every $\varphi \in \Phi_{\mathrm{P}}$, $2\varphi$ is stochastically mixable. The result then follows from Lemma 14, after looking at how each side of the variance condition scales with the change $\varphi \mapsto 2\varphi$. Observe that every $\varphi \in \Phi_{\mathrm{P}}$ is of the form

$$\varphi(y, a) = -(\eta(a) - f(a)) - \log\left(\frac{f(a)}{\eta(a)}\right)^y$$

for some $f \in \mathcal{F}$. Thus, for a fixed $a$, letting $Y \sim P_a$ with $\mathbf{E}[Y] = \eta(a)$, we have

$$\mathbf{E}\exp\{-\varphi(Y, a)\} = \exp\{\eta(a) - f(a)\}\mathbf{E}\left[\left(\frac{f(a)}{\eta(a)}\right)^Y\right].$$

Now, noting that for any $x > 0$ and $y \in [0, 1]$, by convexity, $x^y \leq xy + 1 - y$,

$$\mathbf{E}\left[\left(\frac{f(a)}{\eta(a)}\right)^Y\right] \leq f(a)\frac{\mathbf{E}[Y]}{\eta(a)} + 1 - \mathbf{E}[Y] = 1 + f(a) - \eta(a) \leq \exp\{f(a) - \eta(a)\}.$$

$$(1 + x \leq e^x \text{ for all } x \in \mathbf{R})$$

Combining with the previous display, we have that $\mathbf{E}\exp\{-(2\varphi(Y, a))/2\} \leq 1$. ∎

## C.2 Establishing the triangle condition

To establish the triangle condition, we first sandwich $\Delta$ with an easier-to-work-with quantity:

**Lemma 17.** *For any $p, q \in [0, 1]$,*

$$(\sqrt{p} - \sqrt{q})^2 \leq \Delta(p, q) \leq 2(\sqrt{p} - \sqrt{q})^2$$

*Proof.* If $p, q = 0$ the statement is trivial. Assume that one of $p$ and $q$ is nonzero. Using the algebraic identity $(a - b)(a + b) = a^2 - b^2$, we have

$$(\sqrt{p} + \sqrt{q})^2 (\sqrt{p} - \sqrt{q})^2 = (p - q)^2.$$

Rearranging the above display gives the lower bound:

$$(\sqrt{p} - \sqrt{q})^2 = \frac{(p-q)^2}{(\sqrt{p}+\sqrt{q})^2} \le \frac{(p-q)^2}{p+q} = \Delta(p,q)\,.$$

For the upper bound, note that

$$\Delta(p,q) = \frac{(p-q)^2}{p+q} \le 2\frac{(p-q)^2}{(\sqrt{p}+\sqrt{q})^2} = 2(\sqrt{p}-\sqrt{q})^2\,. \qquad \blacksquare$$

We will also need the following relation between the squared Hellinger distance and Kullback-Leibler divergence, which appears as Equation 7.33 in Polyanskiy and Wu (2025).

**Proposition 18.** *For any two measures $P, Q$ on the same measurable space with densities $p$ and $q$ with respect to some common dominating measure $\mu$,*

$$\mathrm{KL}(Q\|P) \ge \log_2 e \cdot H^2(Q,P) \quad where \quad H^2(Q,P) := \int(\sqrt{p}-\sqrt{q})^2 d\mu\,.$$

We are now ready to prove that the log-loss and Poisson loss functions satisfy the triangle condition.

**Proposition 19** (Log-loss triangle condition)**.** *The expected excess log-loss class $\bar{\Phi}_{\mathrm{X}}$ satisfies the triangle condition with constant $\gamma = 2/\log_2(e)$.*

*Proof.* Let $Q, P$ be Bernoulli distributions with parameters $q \in [0,1]$ and $p \in (0,1)$ respectively, and recall that

$$H^2(Q,P) = (\sqrt{p}-\sqrt{q})^2 + (\sqrt{1-p}-\sqrt{1-q})^2$$

and that

$$\mathrm{KL}(Q\|P) = q\log\frac{q}{p} + (1-q)\log\frac{1-q}{1-p}\,.$$

Using Lemma 17, the definition of $H^2$ and Proposition 18, we have that

$$\Delta(p,q) \le 2(\sqrt{p}-\sqrt{q})^2 \le 2H^2(Q,P) \le (2/\log_2(e))\,\mathrm{KL}(Q\|P)\,.$$

We conclude by observing that for any random variable $Y \in [0,1]$ with mean $q$,

$$\mathbf{E}[\ell_{\mathrm{X}}(Y,p) - \ell_{\mathrm{X}}(Y,q)] = \mathbf{E}\left[Y\log\frac{q}{p} + (1-Y)\log\frac{1-q}{1-p}\right] = \mathrm{KL}(Q\|P)\,. \qquad \blacksquare$$

**Proposition 20** (Poisson loss triangle condition)**.** *The expected excess Poisson loss class $\bar{\Phi}_{\mathrm{P}}$ satisfies the triangle condition with constant $\gamma = 4\sqrt{e}/\log_2(e)$.*

*Proof.* Let $Q, P$ be Poisson distributions with parameters $q \in [0,1]$ and $p \in (0,1]$ respectively, and recall that

$$H^2(Q,P) = 1 - \exp\{-(\sqrt{p}-\sqrt{q})^2/2\} \quad and \quad \mathrm{KL}(Q\|P) = p - q + q\log\frac{q}{p}\,.$$

Observe that for all $x \in [0,1]$, we have the numerical inequality

$$1 - e^{-x/2} \ge x/(2\sqrt{e})\,. \tag{2}$$

Hence,

$$\begin{aligned}
\Delta(p,q) &\le 2(\sqrt{p}-\sqrt{q})^2 & \text{(Lemma 17)} \\
&\le 4\sqrt{e}(1 - \exp\{-(\sqrt{p}-\sqrt{q})^2/2\}) & \text{(Eq. (2))} \\
&= 4\sqrt{e}H^2(Q,P) & \text{(definition of } H^2) \\
&\le \frac{4\sqrt{e}}{\log_2(e)}\,\mathrm{KL}(Q\|P)\,. & \text{(Proposition 18)}
\end{aligned}$$

Now, observe that for any random variable $Y \in [0,1]$ with $\mathbf{E}Y = q$,

$$\mathbf{E}[\ell_{\mathrm{P}}(Y,p) - \ell_{\mathrm{P}}(Y,q)] = \mathbf{E}\left[p - q + Y\log\frac{q}{p}\right] = \mathrm{KL}(Q\|P)\,. \qquad \blacksquare$$

# D  Proof of the cost-sensitive regret bound in the bandit setting, Theorem 1

Recall the following assumptions, and the statement of Theorem 1, which we shall now prove.

**Assumption 2** (Bounded costs). *We have $\cup_{a\in\mathcal{A}} \text{supp } P_a \subset [0,1]$.*

**Assumption 3** (Realisability). *We have that $\eta \in \mathcal{F}$.*

**Definition 1.** Let $\mathcal{F}$ be a model class and $\ell\colon [0,1] \times [0,1] \to \mathbf{R}$ a loss function. For each $f \in \mathcal{F}$, we define the excess loss $\varphi_f\colon [0,1] \times \mathcal{A} \to \mathbf{R}$ and expected excess loss $\bar{\varphi}_f\colon \mathcal{A} \to \mathbf{R}_+$ as

$$\varphi_f(y,a) = \ell(y, f(a)) - \ell(y, \eta(a)) \quad \text{and} \quad \bar{\varphi}_f(a) = \int \varphi_f(\cdot, a) dP_a\,.$$

We will write $\Phi(\mathcal{F}) = \{\varphi_f\colon f \in \mathcal{F}\}$ and $\bar{\Phi}(\mathcal{F}) = \{\bar{\varphi}_f\colon f \in \mathcal{F}\}$ for the respective loss classes.

**Assumption 4** (Loss function assumptions). *There exist constants $b, c, \gamma > 0$ such that for all $(f, a) \in \mathcal{F} \times \mathcal{A}$, letting $Y \sim P_a$, the following three bounds hold:*

$$|\varphi_f(Y, a)| \le b \text{ a.s.}\,, \qquad\qquad\qquad \text{(bounded loss)}$$
$$\text{Var}\,\varphi_f(Y, a) \le c\bar{\varphi}_f(a)\,, \qquad\qquad\quad \text{(variance condition)}$$
$$\Delta(f(a), \eta(a)) \le \gamma\bar{\varphi}_f(a)\,. \qquad\qquad\quad \text{(triangle condition)}$$

**Theorem 1** (Regret bound for $\ell$-UCB in bandits). *Fix $\delta \in (0,1)$, $n \in \mathbf{N}_+$, bandit instance $\mathcal{P}$, model class $\mathcal{F}$ and a loss function $\ell$. Suppose that $(\mathcal{P}, \mathcal{F}, \ell)$ satisfy Assumptions 2 to 4. Let $N_n$ denote the $1/n$-covering number of $\Phi(\mathcal{F})$ with respect to the uniform metric, and for each $t \in \mathbf{N}_+$ let*

$$\beta_t = 5/2 + 15(2b + c)\log(N_n h_t/\delta) \quad \text{where} \quad h_t = e + \log(1 + t)\,.$$

*Define*

$$d_n^\sigma = \dim_{\text{elud}}^\sigma(1/n; \bar{\Phi}(\mathcal{F})) \quad \text{and} \quad \Gamma_n^\sigma = \gamma(1 + (d_n^\sigma + 1)b + 4d_n^\sigma \beta_n \log(1 + nb))\,.$$

*Suppose a learner uses Algorithm 1, $\ell$-UCB, over the course of $n$-many interactions with $\mathcal{P}$, with model class $\mathcal{F}$, loss function $\ell$ and confidence widths $(\beta_t)_{t\ge 1}$. Then, with probability at least $1 - \delta$,*

$$R_n \le \inf_{\sigma \in [1/n, b]} \left\{ 3\sqrt{n\eta(a_\star)\Gamma_n^\sigma} + 6\Gamma_n^\sigma + \left(\frac{4\beta_n}{\sigma} + 1\right) \dim_{\text{elud}}^\sigma(\sigma; \bar{\Phi}(\mathcal{F})) + 1 \right\}\,.$$

*Proof of Theorem 1.* Our proof will rely on Theorem 9, the uniform Bernstein inequality established in Appendix B.

**Validity of confidence sequence** Let $\mathbf{F}$ be the filtration given by $\mathbf{F}_t = \sigma(A_1, Y_1, \ldots, A_t, Y_t, A_{t+1})$ for each $t \in \mathbf{N}$. We apply our uniform Bernstein inequality (Theorem 9) to the $\mathbf{F}$-adapted process $(Y_t, A_t)_{t\in\mathbf{N}_+}$ with the function class $\Phi = \{\varphi_f\colon f \in \mathcal{F}\}$ and the choice $\varepsilon = 1/n$ (the two requirements in Theorem 9 are satisfied due to the boundedness, with $B = 2b$, and variance condition parts of Assumption 4). From this, we conclude the first part of the following proposition:

**Proposition 21.** *There exists an event $\mathcal{E}_\delta$ satisfying $\mathbf{P}(\mathcal{E}_\delta) \ge 1 - \delta$, whereon, for all $f \in \mathcal{F}$ and $t \le n$,*

$$\sum_{i=1}^{t-1} \bar{\varphi}_f(A_i) \le 2\sum_{i=1}^{t-1} \varphi_f(Y_i, A_i) + 2\beta_t\,. \tag{3}$$

*Moreover, on $\mathcal{E}_\delta$,*

$$\eta \in \bigcap_{t\in\mathbf{N}_+} \mathcal{F}_t\,.$$

*Proof.* The first part is immediate. To show that $\mathcal{E}_\delta$, $\eta \in \cap_{t\in\mathbf{N}_+} \mathcal{F}_t$, observe that the left-hand side of Eq. (3) is nonnegative (as ensured by the triangle condition of Assumption 4). From this, we conclude that on $\mathcal{E}_\delta$, for all $t \le n$,

$$0 \le \inf_{\hat{f}\in\mathcal{F}} \sum_{i=1}^{t-1} \varphi_{\hat{f}}(Y_i, A_i) + \beta_t \iff \sum_{i=1}^{t-1} \ell(Y_i, \eta(A_i)) \le \inf_{\hat{f}\in\mathcal{F}} \sum_{i=1}^{t-1} \ell(Y_i, \hat{f}(A_i)) + \beta_t\,.$$

Comparing the right-hand side of the above implication with the form of our confidence set $\mathcal{F}_t$ yields the second conclusion. ∎

**Per-step regret bound**   Bounding the per-step regret will use the following simple inequality for the triangle discrimination, based on an inequality of Ayoub et al. (2024).

**Lemma 22.** *For $x, y, z \geq 0$ with $y \leq z$, we have that $x - z \leq 3\sqrt{z\Delta(x, y)} + 6\Delta(x, y)$.*

**Lemma 23** (Ayoub et al. (2024)). *For $x, z \geq 0$, $z \leq 3x + \Delta(x, z)$.*

*Proof of Lemma 22.*  Observe that

$$x - z \leq x - y \qquad \qquad \text{(by assumption)}$$
$$\leq \sqrt{x + y}\sqrt{\Delta(x, y)} \qquad \qquad \text{(defn. } \Delta(x, y))$$
$$\leq \sqrt{4x + \Delta(x, y)}\sqrt{\Delta(x, y)} \qquad \qquad \text{(Lemma 23)}$$
$$\leq 2\sqrt{x\Delta(x, y)} + \Delta(x, y) \,. \qquad \qquad (4)$$

Hence, applying Young's inequality, we obtain the inequality

$$x \leq 2\sqrt{x\Delta(x, y)} + \Delta(x, y) + z \leq \frac{x}{2} + 3\Delta(x, y) + z \,,$$

which yields that $x \leq 6\Delta(x, y) + 2z$; using this and Eq. (4) gives

$$x - z \leq 2\sqrt{(6\Delta(x, y) + 2z)\, \Delta(x, y)} + \Delta(x, y) \,.$$

We finish by applying $\sqrt{a + b} \leq \sqrt{a} + \sqrt{b}$ for $a, b \geq 0$ in the above, and bounding constants.   ∎

We now apply Lemma 22 to bound per-step regret. For this, note that on $\mathcal{E}_\delta$, for any $t \in \mathbf{N}_+$, by definition of the pair $(f_t, A_t)$ and since $\eta \in \mathcal{F}_t$, we have

$$f_t(A_t) \leq \eta(a_\star) \,.$$

Hence, we may apply Lemma 22 with $x = \eta(A_t)$, $y = f_t(A_t)$ and $z = \eta(a_\star)$ to obtain the bound

$$r_t := \eta(A_t) - \eta(a_\star) \leq 3\sqrt{\eta(a_\star)\Delta(f_t(A_t), \eta(A_t))} + 6\Delta(f_t(A_t), \eta(A_t)) \,. \qquad (5)$$

**Regret decomposition**   Let

$$I_n := \{t \leq n \colon \bar{\varphi}_{f_t}(A_t) \leq \sigma\} \,.$$

Since per-step regret is bounded by 1 (by Assumption 2), for any $n \in \mathbf{N}_+$,

$$R_n = \sum_{t=1}^{n} r_t \leq \sum_{t \in I_n} r_t + \text{card}([n] \setminus I_n) \,.$$

Using Eq. (5), Cauchy-Schwarz, and that $\text{card}\, I_n \leq n$, we have that on $\mathcal{E}_\delta$,

$$\sum_{t \in I_n} r_t \leq 3\sqrt{n\eta(a_\star) \sum_{t \in I_n} \Delta(f_t(A_t), \eta(A_t))} + 6\sum_{t \in I_n} \Delta(f_t(A_t), \eta(A_t))$$

Now, observe that by the triangle condition of Assumption 4,

$$\sum_{t \in I_n} \Delta(f_t(A_t), \eta(A_t)) \leq \gamma \sum_{t \in I_n} \bar{\varphi}_{f_t}(A_t) \,.$$

Therefore, we need only upper bound the two quantities

$$\text{card}([n] \setminus I_n) \quad \text{and} \quad \sum_{t \in I_n} \bar{\varphi}_{f_t}(A_t) \,.$$

For this, we will use the following lemma, proven after the conclusion of the current proof.

**Lemma 24.** *Fix $B, \beta > 0$ and $0 < \sigma \leq B$. Let $\mathcal{X}$ be a set and $\Psi$ a set of $[0, B]$-valued functions on $\mathcal{X}$. Suppose sequences $(x_1, \ldots, x_n) \in \mathcal{X}^n$ and $(\psi_1, \ldots, \psi_n) \in \Psi^n$ satisfy*

$$\sum_{i=1}^{t-1} \psi_t(x_i) \leq \beta \qquad \text{for all } t \in [n].$$

*Then the following hold.*

*1. For every $\varepsilon \in (0, \sigma]$ and every $t \in [n]$,*

$$\sum_{i=1}^{t} \mathbf{1}\{\psi_i(x_i) > \varepsilon\} \leq \left(\frac{\beta}{\varepsilon} + 1\right) \mathrm{dim}_{\mathrm{elud}}^{\sigma}(\varepsilon; \Psi) + 1.$$

*2. If $\psi_t(x_t) \leq \sigma$ for all $t \in [n]$, then for every $\omega \in (0, \sigma]$ and every $t \in [n]$,*

$$\sum_{i=1}^{t} \psi_i(x_i) \leq \left(B + \beta \log\left(1 + \frac{B}{\omega}\right)\right) \mathrm{dim}_{\mathrm{elud}}^{\sigma}(\omega; \Psi) + B + t\omega.$$

In particular, we will apply the above lemma with the sequence given by

$$\psi_i = \bar{\varphi}_{f_i} \quad \text{and} \quad x_i = A_i, \qquad i = 1, \ldots, t.$$

The following proposition confirms that this sequence satisfies the condition with $\beta = 4\beta_t$.

**Proposition 25.** *On the event $\mathcal{E}_\delta$ of Proposition 21, we have that for all $t \in \mathbf{N}_+$,*

$$\sum_{i=1}^{t-1} \bar{\varphi}_{f_t}(A_i) \leq 4\beta_t.$$

*Proof of Proposition 25.* On $\mathcal{E}_\delta$, for any $t \in \mathbf{N}_+$,

$$
\begin{aligned}
\sum_{i=1}^{t-1} \bar{\varphi}_{f_t}(A_i) &\leq 2\left[\sum_{i=1}^{t-1} \varphi_{f_t}(Y_i, A_i) + \beta_t\right] && \text{(on } \mathcal{E}_\delta \text{ Eq. (3) holds)} \\
&= 2\left[\sum_{i=1}^{t-1} \ell(Y_i, f_t(A_i)) - \sum_{i=1}^{t-1} \ell(Y_i, \eta(A_i)) + \beta_t\right] \\
&\leq 2\left[\sum_{i=1}^{t-1} \ell(Y_i, f_t(A_i)) - \inf_{\hat{f} \in \mathcal{F}} \sum_{i=1}^{t-1} \ell(Y_i, \hat{f}(A_i)) + \beta_t\right] && (\eta \in \mathcal{F}) \\
&\leq 4\beta_t. && (f_t \in \mathcal{F}_t \text{ and the definition of } \mathcal{F}_t)
\end{aligned}
$$

∎

With that, by the first part of Lemma 24 and using that $\beta_t \leq \beta_n$,

$$\mathrm{card}([n] \setminus I_n) = \sum_{t=1}^{n} \mathbf{1}\{|\psi_t(x_t)| > \sigma\} \leq \left(\frac{4\beta_n}{\sigma} + 1\right) \mathrm{dim}_{\mathrm{elud}}^{\sigma}(\sigma; \bar{\Phi}(\mathcal{F})) + 1.$$

For the sum of the regret along the good subsequence $I_n$, first write

$$I_n = \{\tau_1 < \tau_2 < \cdots < \tau_m\}.$$

We now pass to this subsequence, writing

$$\tilde{x}_j = A_{\tau_j}, \qquad \tilde{\psi}_j = \bar{\varphi}_{f_{\tau_j}}, \qquad j \in [m].$$

Since $\tau_j \in I_n$, we have that the condition

$$\tilde{\psi}_j(\tilde{x}_j) \leq \sigma \qquad \text{for all } j \in [m]$$

holds for this subsequence. Moreover, since $I_n \subset [n]$, we can again take $\beta = 4\beta_t \leq 4\beta_n$. Therefore, by the second part of Lemma 24 with $\omega = 1/n$,

$$\sum_{t \in I_n} \bar{\varphi}_{f_t}(A_t) \leq (4\beta_n \log(1 + bn) + b)\mathrm{dim}_{\mathrm{elud}}^{\sigma}(1/n; \Psi) + b + 1.$$

Combining the established inequalities and taking infimum over $\sigma \in [1/n, b]$ concludes the proof. ∎

### D.1 Proof of Lemma 24 (control by localised eluder dimension)

*Proof of Lemma 24. Proof of part 1.* Fix $\varepsilon \in (0, \sigma]$, $t \in [n]$, and write

$$d = \dim^{\sigma}_{\mathrm{elud}}(\varepsilon; \Psi).$$

If $d = 0$, then by definition of $\dim^{\sigma}_{\mathrm{elud}}(\varepsilon; \Psi)$ there is no $x \in \mathcal{X}$ and no $\psi \in \Psi$ such that $\psi(x) > \varepsilon$, so the claim is trivial.

Assume now that $d \geq 1$, and let $z_1, \ldots, z_m$ be the subsequence of $x_1, \ldots, x_t$ consisting of those $x_i$ for which $\psi_i(x_i) > \varepsilon$. The proof of Claims 1 and 2, and hence of Lemma 23, in Appendix D.2 of Liu et al. (2022) applies verbatim here: the argument only uses $\varepsilon$-independence at the fixed scale $\varepsilon$, so one may replace the global eluder dimension there by the localised quantity $\dim^{\sigma}_{\mathrm{elud}}(\varepsilon; \Psi)$. Consequently, there exists some $j \in [m]$ such that $z_j$ is $\varepsilon$-dependent on at least

$$\left\lfloor \frac{m-1}{d} \right\rfloor$$

pairwise disjoint subsequences of $(z_1, \ldots, z_{j-1})$, while at the same time any such point can be $\varepsilon$-dependent on at most $\beta/\varepsilon$ pairwise disjoint subsequences. Hence

$$\left\lfloor \frac{m-1}{d} \right\rfloor \leq \frac{\beta}{\varepsilon},$$

and therefore

$$m \leq \left( \frac{\beta}{\varepsilon} + 1 \right) d + 1.$$

This is exactly the desired bound.

*Proof of part 2.* Assume that $\psi_i(x_i) \leq \sigma$ for all $i \in [n]$, and fix $\omega \in (0, \sigma]$ and $t \in [n]$. Let

$$d = \dim^{\sigma}_{\mathrm{elud}}(\omega; \Psi).$$

Since each $\psi_i(x_i) \in [0, \sigma]$, we have

$$\sum_{i=1}^{t} \psi_i(x_i) = \sum_{i=1}^{t} \int_0^{\sigma} \mathbf{1}\{\psi_i(x_i) > y\} \, dy$$

$$\leq t\omega + \int_{\omega}^{\sigma} \sum_{i=1}^{t} \mathbf{1}\{\psi_i(x_i) > y\} \, dy.$$

Now fix $y \in [\omega, \sigma]$. By part 1 and monotonicity of $\nu \mapsto \dim^{\sigma}_{\mathrm{elud}}(\nu; \Psi)$,

$$\sum_{i=1}^{t} \mathbf{1}\{\psi_i(x_i) > y\} \leq \left( \frac{\beta}{y} + 1 \right) \dim^{\sigma}_{\mathrm{elud}}(y; \Psi) + 1 \leq \left( \frac{\beta}{y} + 1 \right) d + 1.$$

Substituting this into the previous display yields

$$\sum_{i=1}^{t} \psi_i(x_i) \leq t\omega + \int_{\omega}^{\sigma} \left( \left( \frac{\beta}{y} + 1 \right) d + 1 \right) dy$$

$$= t\omega + \left( \beta \log \frac{\sigma}{\omega} + \sigma - \omega \right) d + \sigma - \omega$$

$$\leq t\omega + \left( \beta \log \left( 1 + \frac{B}{\omega} \right) + B \right) d + B,$$

where in the last step we used $\sigma \leq B$. ∎

# E  Proof of RL cost-sensitive regret bound, Theorem 6

Recall the following assumptions, and the statement of Theorem 6, which we shall now prove.

**Assumption 5.** *Costs are nonnegative and sum to at most one over each episode.*

**Assumption 6** (Realisability)**.** *We assume that $q^\star \in \mathcal{F}$.*

**Assumption 7** (Generalised completeness)**.** *We assume that $\mathcal{TF} \subset \mathcal{G}$.*

**Definition 3.** For any $f \in \mathcal{F}$, $h \in [H]$, $x \in \mathcal{S} \times \mathcal{A}$ and $s' \in \mathcal{S}$, we let

$$y_h^f(x, s') = 1 \wedge (c_h(x) + f_{h+1}^{\wedge}(s'))$$

be the response under the model $f$. With the same symbols, we define the excess Bellman loss function

$$\varphi_h^f(x, s') = \ell(y_h^f(x, s'), f_h(x)) - \ell(y_h^f(x, s'), (\mathcal{T}f)_h(x)),$$

and the expected excess Bellman loss function

$$\bar{\varphi}_h^f(x) = \int \varphi_h^f(x, s') \, P_h(ds' \mid x).$$

**Assumption 8** (RL loss function assumptions)**.** *There exist constants $b, c, \gamma > 0$ such that for all $f \in \mathcal{F}$, $h \in [H]$, $x \in \mathcal{S} \times \mathcal{A}$, $S' \sim P_h(x)$, the following hold:*

$$|\varphi_h^f(x, S')| \leq b \text{ a.s.}, \tag{RL boundedness}$$
$$\operatorname{Var} \varphi_h^f(x, S') \leq c\bar{\varphi}_h^f(x), \tag{RL variance condition}$$
$$\Delta(f_h(x), (\mathcal{T}f)_h(x)) \leq \gamma\bar{\varphi}_h^f(x). \tag{RL triangle condition}$$

**Theorem 6.** *Fix $\delta \in (0, 1)$, $n \in \mathbf{N}_+$, MDP $M$, model classes $\mathcal{F}$ and $\mathcal{G}$ and a loss function $\ell$. Suppose that $(M, \mathcal{F}, \mathcal{G}, \ell)$ satisfy Assumptions 6 to 8. Let*

$$\Phi_{\mathrm{RL}}(\mathcal{F}) = \{\varphi_h^f \colon h \in [H], f \in \mathcal{F}\},$$

*let $N_n$ be the $1/n$-covering number of $\Phi_{\mathrm{RL}}(\mathcal{F})$ with respect to the uniform metric, let $h_t = e + \log(1 + t)$ for each $t \in [n]$, and let*

$$\beta_t = 5/2 + 15(3b + c) \log(HN_n h_t/\delta), \qquad t \in \mathbf{N}_+.$$

*For each $f \in \mathcal{F}$ and $h \in [H]$, let $\mu_h^f$ denote the stage-$h$ state-action occupancy measure induced by following the policy $\pi^f$, and define*

$$\Psi_h(\mathcal{F}) = \left\{\nu \mapsto \int \bar{\varphi}_h^g \, d\nu \colon g \in \mathcal{F}\right\},$$

*viewed as a class of $[0, b]$-valued functions on $\{\mu_h^f \colon f \in \mathcal{F}\}$. For $\sigma \in [1/n, b]$, define*

$$d_n^\sigma = \sum_{h=1}^{H} \dim_{\mathrm{elud}}^\sigma(1/n; \Psi_h(\mathcal{F})) \quad \text{and} \quad \tilde{d}_n^\sigma = \sum_{h=1}^{H} \dim_{\mathrm{elud}}^\sigma(\sigma; \Psi_h(\mathcal{F})),$$

*and*

$$\Gamma_n^\sigma = \gamma\big[H + (d_n^\sigma + H)b + 4d_n^\sigma\beta_n \log(1 + nb)\big].$$

*Suppose a learner uses Algorithm 2, $\ell$-GOLF, over the course of $n$-many episodes with $M$, model classes $\mathcal{F}$ and $\mathcal{G}$, loss function $\ell$ and confidence widths $(\beta_t)_{t \in \mathbf{N}_+}$. Then, with probability at least $1 - \delta$,*

$$R_n \leq \inf_{\sigma \in [1/n, b]} \left\{5\sqrt{Hnv_1^\star(s_1)\Gamma_n^\sigma} + 12H\Gamma_n^\sigma + \left(\frac{4\beta_n}{\sigma} + 1\right)\tilde{d}_n^\sigma + H\right\}.$$

Within the upcoming proofs, we will use the shorthand

$$X_h^t = (S_h^t, A_h^t) \qquad \text{and} \qquad \mu_h^t = \mu_h^{f^t}.$$

**Proposition 26.** *There exists an event $\mathcal{E}_\delta$ satisfying $\mathbf{P}(\mathcal{E}_\delta) \geq 1 - \delta$, whereon, for all $f \in \mathcal{F}$, $h \in [H]$ and $t \in \mathbf{N}_+$,*

$$\sum_{i=1}^{t-1} \int \bar{\varphi}_h^f \, d\mu_h^i \leq 2 \sum_{i=1}^{t-1} \varphi_h^f(X_h^i, S_{h+1}^i) + 2\beta_t \,. \tag{6}$$

*Moreover, deterministically,*

$$q^\star \in \bigcap_{t \leq n} \mathcal{F}^t \,.$$

*Proof.* Let $(\mathcal{H}_t)_{t \in \mathbf{N}}$ be the filtration generated by the first $t$ complete episodes. For each fixed $h \in [H]$, consider the $\mathcal{H}$-adapted process

$$Z_i^h = (X_h^i, S_{h+1}^i), \qquad i \in \mathbf{N}_+,$$

together with the function class $\Phi_h(\mathcal{F}) = \{\varphi_h^f \colon f \in \mathcal{F}\}$. By definition of $\mu_h^i$,

$$\mathbf{E}[\varphi_h^f(Z_i^h) \mid \mathcal{H}_{i-1}] = \int \bar{\varphi}_h^f \, d\mu_h^i.$$

Also, since $|\varphi_h^f| \leq b$ and $0 \leq \bar{\varphi}_h^f \leq b$,

$$\left| \mathbf{E}[\varphi_h^f(Z_i^h) \mid \mathcal{H}_{i-1}] - \varphi_h^f(Z_i^h) \right| \leq 2b.$$

Moreover, by the law of total variance,

$$\begin{aligned} \mathrm{Var}(\varphi_h^f(Z_i^h) \mid \mathcal{H}_{i-1}) &= \mathbf{E}\left[ \mathrm{Var}(\varphi_h^f(Z_i^h) \mid X_h^i, \mathcal{H}_{i-1}) \mid \mathcal{H}_{i-1} \right] + \mathrm{Var}(\bar{\varphi}_h^f(X_h^i) \mid \mathcal{H}_{i-1}) \\ &\leq c \int \bar{\varphi}_h^f \, d\mu_h^i + b \int \bar{\varphi}_h^f \, d\mu_h^i \\ &= (b+c) \int \bar{\varphi}_h^f \, d\mu_h^i, \end{aligned}$$

where the second term is bounded using $0 \leq \bar{\varphi}_h^f \leq b$. Since the $1/n$-covering number of $\Phi_h(\mathcal{F})$ is at most $N_n$, applying Theorem 9 with confidence parameter $\delta/H$ and taking a union bound over $h \in [H]$ yields the first claim.

For the second claim, note that by Assumptions 6 and 7 and Eq. (1), we have $q^\star \in \mathcal{F}$ and $(\mathcal{T}q^\star)_h = q_h^\star \in \mathcal{G}_h$ for every $h \in [H]$. Hence, for all $t \in \mathbf{N}_+$ and $h \in [H]$,

$$L_h^{t-1}(q_{h+1}^\star, q_h^\star) = L_h^{t-1}(q_{h+1}^\star, (\mathcal{T}q^\star)_h) \leq \inf_{g \in \mathcal{G}_h} L_h^{t-1}(q_{h+1}^\star, g) + \beta_t,$$

so $q^\star \in \mathcal{F}^t$. ∎

**Lemma 27** (Contraction lemma). *Let $f \in \mathcal{F}$, let $\pi := \pi^f$, and let $v = v^\pi$. Then*

$$\sqrt{\Delta(f_1^\wedge(s_1), v_1(s_1))} \leq \sqrt{2} \sum_{h=1}^{H} \sqrt{\mathbf{E}_\pi[\Delta(f_h(X_h), (\mathcal{T}f)_h(X_h))]},$$

*where $\mathbf{E}_\pi$ denotes the expectation over the state-action pairs $(X_h)_{h=1}^{H}$ resulting from following the policy $\pi$ in the MDP $M$.*

**Lemma 28.** *On $\mathcal{E}_\delta$, for all $t \in \mathbf{N}_+$ and $h \in [H]$,*

$$\sum_{i=1}^{t-1} \int \bar{\varphi}_h^{f^t} \, d\mu_h^i \leq 4\beta_t \,.$$

*Proof.* By Proposition 26, on $\mathcal{E}_\delta$,

$$\sum_{i=1}^{t-1} \int \bar{\varphi}_h^{f^t} \, d\mu_h^i \leq 2 \sum_{i=1}^{t-1} \varphi_h^{f^t}(X_h^i, S_{h+1}^i) + 2\beta_t.$$

Now observe that

$$\sum_{i=1}^{t-1} \varphi_h^{f^t}(X_h^i, S_{h+1}^i) = L_h^{t-1}(f_{h+1}^t, f_h^t) - L_h^{t-1}(f_{h+1}^t, (\mathcal{T}f^t)_h)$$

$$\leq \beta_t,$$

because $f^t \in \mathcal{F}^t$ and $(\mathcal{T}f^t)_h \in \mathcal{G}_h$ by Assumption 7. Combining the two displays proves the claim. ∎

*Proof of Theorem 6.* For each $t \in \mathbf{N}_+$, write

$$r_t = v_1^t(s_1) - v_1^\star(s_1).$$

Since $q^\star \in \mathcal{F}^t$ by Proposition 26 and $f^t$ is chosen optimistically over $\mathcal{F}^t$, we have

$$f_1^{t\wedge}(s_1) \leq q_1^{\star\wedge}(s_1) = v_1^\star(s_1).$$

Hence, by Lemma 22,

$$r_t \leq 3\sqrt{v_1^\star(s_1)\Delta(v_1^t(s_1), f_1^{t\wedge}(s_1))} + 6\Delta(v_1^t(s_1), f_1^{t\wedge}(s_1)).$$

Fix $\sigma \in [1/n, b]$, and let

$$I_n^\sigma = \left\{ t \leq n \colon \int \bar{\varphi}_h^{f^t} d\mu_h^t \leq \sigma, \ \forall h \in [H] \right\}.$$

Since the cumulative cost in every episode is at most 1 by Assumption 5, we have

$$R_n \leq \sum_{t \in I_n^\sigma} r_t + \mathrm{card}([n] \setminus I_n^\sigma).$$

For $t \in I_n^\sigma$ and $h \in [H]$, let

$$\Delta_{t,h} = \Delta(f_h^t(X_h), (\mathcal{T}f^t)_h(X_h)),$$

where the expectation is taken with respect to the trajectory induced by $\pi^t = \pi^{f^t}$. By Lemma 27, the symmetry of $\Delta$, and the bounds $3\sqrt{2} < 5$ and $6 \cdot 2 = 12$,

$$\sum_{t \in I_n^\sigma} r_t \leq 5 \sum_{t \in I_n^\sigma} \sum_{h=1}^{H} \sqrt{v_1^\star(s_1)\mathbf{E}_{\pi^t}[\Delta_{t,h}]} + 12 \sum_{t \in I_n^\sigma} \left\{ \sum_{h=1}^{H} \sqrt{\mathbf{E}_{\pi^t}[\Delta_{t,h}]} \right\}^2$$

$$\leq 5\sqrt{Hnv_1^\star(s_1) \sum_{t \in I_n^\sigma} \sum_{h=1}^{H} \mathbf{E}_{\pi^t}[\Delta_{t,h}]} + 12H \sum_{t \in I_n^\sigma} \sum_{h=1}^{H} \mathbf{E}_{\pi^t}[\Delta_{t,h}],$$

where we used Cauchy–Schwarz and $\mathrm{card}(I_n^\sigma) \leq n$.

Now, by the RL triangle condition in Assumption 8,

$$\sum_{t \in I_n^\sigma} \sum_{h=1}^{H} \mathbf{E}_{\pi^t}[\Delta_{t,h}] \leq \gamma \sum_{t \in I_n^\sigma} \sum_{h=1}^{H} \int \bar{\varphi}_h^{f^t} d\mu_h^t. \tag{7}$$

For each $h \in [H]$ and $t \in \mathbf{N}_+$, define $\psi_h^t \in \Psi_h(\mathcal{F})$ by

$$\psi_h^t(\nu) = \int \bar{\varphi}_h^{f^t} d\nu.$$

By Lemma 28, on $\mathcal{E}_\delta$,

$$\sum_{i=1}^{t-1} \psi_h^t(\mu_h^i) \leq 4\beta_t \leq 4\beta_n \qquad \forall t \in \mathbf{N}_+, \ \forall h \in [H].$$

We now apply the two parts of Lemma 24 stage-wise, with

$$X = \{\mu_h^f : f \in \mathcal{F}\}, \qquad \Psi = \Psi_h(\mathcal{F}), \qquad x_i = \mu_h^i, \qquad \psi_i = \psi_h^i, \qquad B = b, \qquad \beta = 4\beta_n.$$

First, the bad episodes satisfy

$$\begin{aligned}
\mathrm{card}([n] \setminus I_n^\sigma) &\leq \sum_{h=1}^H \mathrm{card}\{t \leq n : \psi_h^t(\mu_h^t) > \sigma\} \\
&\leq \sum_{h=1}^H \left\{ \left( \frac{4\beta_n}{\sigma} + 1 \right) \mathrm{dim}_{\mathrm{elud}}^\sigma(\sigma; \Psi_h(\mathcal{F})) + 1 \right\} \\
&= \left( \frac{4\beta_n}{\sigma} + 1 \right) \tilde{d}_n^\sigma + H.
\end{aligned}$$

Next, write $I_n^\sigma = \{\tau_1 < \cdots < \tau_m\}$. Since $\psi_h^{\tau_j}(\mu_h^{\tau_j}) \leq \sigma$ for all $j \in [m]$, and since by nonnegativity

$$\sum_{\ell=1}^{j-1} \psi_h^{\tau_j}(\mu_h^{\tau_\ell}) \leq \sum_{i=1}^{\tau_j - 1} \psi_h^{\tau_j}(\mu_h^i) \leq 4\beta_n,$$

the second part of Lemma 24 gives, for each $h \in [H]$,

$$\begin{aligned}
\sum_{t \in I_n^\sigma} \int \bar{\varphi}_h^{f^t} \, d\mu_h^t &= \sum_{j=1}^m \psi_h^{\tau_j}(\mu_h^{\tau_j}) \\
&\leq (b + 4\beta_n \log(1 + bn)) \, \mathrm{dim}_{\mathrm{elud}}^\sigma(1/n; \Psi_h(\mathcal{F})) + b + m/n \\
&\leq (b + 4\beta_n \log(1 + bn)) \, \mathrm{dim}_{\mathrm{elud}}^\sigma(1/n; \Psi_h(\mathcal{F})) + b + 1.
\end{aligned}$$

Summing over $h$ yields

$$\begin{aligned}
\sum_{t \in I_n^\sigma} \sum_{h=1}^H \int \bar{\varphi}_h^{f^t} \, d\mu_h^t &\leq (b + 4\beta_n \log(1 + bn)) \, d_n^\sigma + H(b+1) \\
&= \Gamma_n^\sigma / \gamma.
\end{aligned}$$

Combining the last display with (7) and the earlier regret decomposition, we conclude that on $\mathcal{E}_\delta$,

$$R_n \leq 5\sqrt{Hn v_1^\star(s_1)\Gamma_n^\sigma} + 12H\Gamma_n^\sigma + \left( \frac{4\beta_n}{\sigma} + 1 \right) \tilde{d}_n^\sigma + H.$$

Taking the infimum over $\sigma \in [1/n, b]$ completes the proof. ∎

We now prove the contraction lemma, Lemma 27. We will need the following simple result.

**Lemma 29.** *For $x, y \geq 0$, the map $(x, y) \mapsto (\sqrt{x} - \sqrt{y})^2$ is jointly convex.*

*Proof.* Since

$$(\sqrt{x} - \sqrt{y})^2 = x + y - 2\sqrt{xy},$$

and $(x, y) \mapsto -\sqrt{xy}$ is jointly convex on $\mathbf{R}_+^2$, the claim follows. ∎

*Proof of Lemma 27.* For each $h \in [H]$, let $\mu_h$ denote the law of $X_h = (S_h, \pi_h(S_h))$ under $\pi$, and, by abuse of notation, identify $v_h$ with the function $(s, a) \mapsto v_h(s)$ on $\mathcal{S} \times \mathcal{A}$.

By Lemma 17,

$$\sqrt{\Delta(f_1^\wedge(s_1), v_1(s_1))} \leq \sqrt{2} \, \|\sqrt{f_1} - \sqrt{v_1}\|_{\mu_1}.$$

We claim that for every $h \in [H]$,

$$\|\sqrt{f_h} - \sqrt{v_h}\|_{\mu_h} \leq \|\sqrt{f_h} - \sqrt{(\mathcal{T}f)_h}\|_{\mu_h} + \|\sqrt{f_{h+1}} - \sqrt{v_{h+1}}\|_{\mu_{h+1}}. \tag{8}$$

Granting this for the moment, unrolling (8) over $h = 1, \ldots, H$ and using $f_{H+1} = v_{H+1} = 0$ gives

$$\|\sqrt{f_1} - \sqrt{v_1}\|_{\mu_1} \leq \sum_{h=1}^{H} \|\sqrt{f_h} - \sqrt{(\mathcal{T}f)_h}\|_{\mu_h}.$$

The lower bound in Lemma 17 now implies

$$\|\sqrt{f_h} - \sqrt{(\mathcal{T}f)_h}\|_{\mu_h} \leq \sqrt{\mathbf{E}_\pi[\Delta(f_h(X_h), (\mathcal{T}f)_h(X_h))]},$$

and the lemma follows.

It remains to prove (8). Fix $h \in [H]$. By the triangle inequality,

$$\|\sqrt{f_h} - \sqrt{v_h}\|_{\mu_h} \leq \|\sqrt{f_h} - \sqrt{(\mathcal{T}f)_h}\|_{\mu_h} + \|\sqrt{(\mathcal{T}f)_h} - \sqrt{v_h}\|_{\mu_h}.$$

For the second term, write

$$A_h = f_{h+1}^{\wedge}(S_{h+1}) \qquad \text{and} \qquad B_h = v_{h+1}(S_{h+1}).$$

Then

$$(\mathcal{T}f)_h(X_h) = c_h(X_h) + \mathbf{E}_\pi[A_h \mid X_h] \qquad \text{and} \qquad v_h(S_h) = c_h(X_h) + \mathbf{E}_\pi[B_h \mid X_h].$$

Using that $(\sqrt{c+a} - \sqrt{c+b})^2 \leq (\sqrt{a} - \sqrt{b})^2$ for $c, a, b \geq 0$, we obtain

$$\|\sqrt{(\mathcal{T}f)_h} - \sqrt{v_h}\|_{\mu_h}^2 \leq \mathbf{E}_\pi\left[\left(\sqrt{\mathbf{E}_\pi[A_h \mid X_h]} - \sqrt{\mathbf{E}_\pi[B_h \mid X_h]}\right)^2\right].$$

Since the map $(x, y) \mapsto (\sqrt{x} - \sqrt{y})^2$ is jointly convex by Lemma 29, Jensen's inequality gives

$$\mathbf{E}_\pi\left[\left(\sqrt{\mathbf{E}_\pi[A_h \mid X_h]} - \sqrt{\mathbf{E}_\pi[B_h \mid X_h]}\right)^2\right] \leq \mathbf{E}_\pi\left[\mathbf{E}_\pi\left[(\sqrt{A_h} - \sqrt{B_h})^2 \mid X_h\right]\right]$$

$$= \mathbf{E}_\pi\left[(\sqrt{A_h} - \sqrt{B_h})^2\right].$$

Finally, because $\pi = \pi^f$ is greedy with respect to $f$, we have $A_h = f_{h+1}(X_{h+1})$ almost surely. Therefore

$$\mathbf{E}_\pi\left[(\sqrt{A_h} - \sqrt{B_h})^2\right] = \|\sqrt{f_{h+1}} - \sqrt{v_{h+1}}\|_{\mu_{h+1}}^2,$$

which proves (8). ∎

# F On self-concordant GLMs with compatible losses

We restate our compatible GLM assumption for convenience.

**Assumption 1.** *We make the following assumptions:*

$$
\begin{array}{rll}
 & \mathcal{A} \subset \mathbf{B}_2^d & \text{(action set bound)} \\
(\exists S > 0) & \Theta \subset S\mathbf{B}_2^d & \text{(parameter set bound)} \\
(\forall (a, \theta) \in \mathcal{A} \times \Theta) & \langle a, \theta \rangle \in U & \text{(valid domain)} \\
(\exists L > 0, \forall u, u' \in U) & |\mu(u) - \mu(u')| \leq L|u - u'| & (L\text{-Lipschitz link}) \\
(\exists M \geq 1, \forall u \in U^\circ) & |\ddot{\mu}(u)| \leq M\dot{\mu}(u) & (M\text{-self-concordant link}) \\
(\exists 1 \leq \kappa < \infty) & \kappa \geq \sup_{u \in U^\circ} 1/\dot{\mu}(u) & \text{(link derivative lower bound)} \\
(\forall y \in [0, 1], \forall u \in U) & \partial_u \ell(y, \mu(u)) = \mu(u) - y\,. & \text{(link and loss are compatible)}
\end{array}
$$

In this section, we prove that the following hold under Assumption 1:

1. the excess loss class $\Phi(\mathrm{GLM}(\mu, \Theta))$ is uniformly bounded and admits a small uniform cover
2. the expected excess loss class $\bar{\Phi}(\mathrm{GLM}(\mu, \Theta))$ has a small localised eluder dimension

These results combined with Theorem 1 yield Proposition 5.

We will use the following lemma repeatedly.

**Lemma 30.** *Fix some $(y, a) \in [0, 1] \times \mathbf{B}_2^d$ and let $h(\theta) = \ell(y, \mu(\langle a, \theta \rangle))$. Let $\theta, \theta' \in \mathbf{R}^d$ and write $\theta(t) = t\theta + (1 - t)\theta'$. Then,*

$$
h(\theta) - h(\theta') = \int_0^1 \partial_t h(\theta(t)) dt = \partial_t h(\theta') + \int_0^1 (1 - t)\partial_t^2 h(\theta(t)) dt\,,
$$

*where*

$$
\partial_t h(\theta(t)) = (\mu(\langle a, \theta(t) \rangle) - y)\langle a, \theta - \theta' \rangle\,, \quad \text{and}
$$
$$
\partial_t^2 h(\theta(t)) = \dot{\mu}(\langle a, \theta(t) \rangle)\langle aa^\mathsf{T}(\theta - \theta'), \theta - \theta' \rangle\,.
$$

*Proof sketch.* The proof follows from the fundamental theorem of calculus for the first equality, and then a Taylor expansion followed by another application of the fundamental theorem of calculus for the second equality. The absolute continuity requisite for the fundamental theorem of calculus is ensured by the $L$-Lipschitz continuity of the link function for the first application, and by the second derivative of the loss being bounded, which may be seen from its form, combined with $L$ being an upper bound on $\dot{\mu}(u)$ for all $u \in U^\circ$. ∎

## F.1 Boundedness of excess losses & covering number bound

We first establish the boundedness of the excess risk class with $b = 4S$, which is implied from the following proposition together with our realisability assumption:

**Lemma 31.** *Let $(\Theta, \mathcal{A}, \mu, \ell)$ be compatible according to Assumption 1. Then, for all $\theta, \theta' \in \Theta$ and $(y, a) \in [0, 1] \times \mathcal{A}$,*

$$
|\ell(y, \mu(\langle a, \theta \rangle)) - \ell(y, \mu(\langle a, \theta' \rangle))| \leq 2\|\theta - \theta'\| \leq 4S\,.
$$

*Proof.* Let $\theta(t) = t\theta + (1 - t)\theta'$ and note that for any $(y, a) \in [0, 1] \times \mathcal{A}$, by Lemma 30,

$$
\begin{aligned}
|\ell(y, \mu(\langle a, \theta \rangle)) - \ell(y, \mu(\langle a, \theta' \rangle))| &= |\int_0^1 (\mu(\langle a, \theta(t) \rangle) - y)\langle a, \theta - \theta' \rangle dt| \\
&\leq |\int_0^1 (\mu(\langle a, \theta(t) \rangle) - y) dt| \, |\langle a, \theta - \theta' \rangle| \\
&\leq 2\|\theta - \theta'\|\,. \quad\quad\quad\quad\quad\quad\quad\quad \blacksquare
\end{aligned}
$$

Now we establish a bound on the corresponding uniform covering number:

**Proposition 32.** *Under Assumption 1, the $\varepsilon$-covering number of $\Phi(\mathrm{GLM}(\mu, \Theta))$ with respect to the uniform norm is upper bounded by $(1 + 8S/\varepsilon)^d$.*

*Proof.* Write $\mathcal{F} = \mathrm{GLM}(\mu, \Theta)$. Let $\theta_\star \in \Theta$ be such that $\eta(a) = \mu(\langle a, \theta_\star \rangle)$ (such a parameter exists by Assumption 3, realisability) and define the map $\rho \colon \Theta \to \Phi(\mathcal{F})$ as that taking each $\theta \in \Theta$ to the map

$$(y, a) \mapsto \ell(y, \mu(\langle a, \theta \rangle)) - \ell(y, \eta(a))\,.$$

Observe that $\Phi(\mathcal{F}) = \rho(\Theta)$, and that since for any $\theta_0, \theta_1 \in \Theta$,

$$\|\rho(\theta_0) - \rho(\theta_1)\|_\infty = \sup_{(y,a) \in [0,1] \times \mathcal{A}} |\ell(y, \mu(\langle a, \theta_0 \rangle)) - \ell(y, \mu(\langle a, \theta_1 \rangle))|\,,$$

we have by Lemma 31 that $\rho$ is 2-Lipschitz as a map from $(\Theta, \|\cdot\|_2) \to (\mathcal{L}(\mathcal{F}), \|\cdot\|_\infty)$. Now, if $\mathcal{C}_{\varepsilon/2}$ is an $\varepsilon/2$-cover of $(S\mathbf{B}_2^d, \|\cdot\|_2)$, then by said 2-Lipschitzness, $\rho(\mathcal{C}_{\varepsilon/2})$ is an $\varepsilon$-external-cover of $(\Phi(\mathcal{F}), \|\cdot\|_\infty)$. Finally, the 2-norm $\varepsilon/4$-covering number of $S\mathbf{B}_2^d$ is an upper bound on the $\varepsilon/2$-covering number of $\Theta$ (Vershynin, 2018, Exercise 4.2.9), and the former quantity is at most $(1 + 8S/\varepsilon)^d$ (Vershynin, 2018, Corollary 4.2.13). ∎

## F.2 The localised eluder dimension

In the following, we associate with each function $f_t$ selected by the $\ell$-UCB algorithm a parameter $\theta_t \in \Theta$ such that $f_t(\cdot) = \mu(\langle \cdot, \theta_t \rangle)$.

By realisability, we can write any $\bar{\varphi} \in \bar{\Phi}(\mathrm{GLM}(\mu, \Theta))$ in the form

$$\bar{\varphi}(a) = \int \ell(y, \mu(\langle a, \theta \rangle)) - \ell(y, \mu(\langle a, \theta_\star \rangle)) P_a(dy) =: \bar{\varphi}(a, \theta) \quad \text{for some } \theta, \theta_\star \in S\mathbf{B}_2^d\,.$$

**Lemma 33.** *For any $\theta \in \mathbf{R}^d$, letting $\theta(t) = t\theta + (1 - t)\theta_\star$ for $t \in [0, 1]$, we have that*

$$\bar{\varphi}(a, \theta) = \|\theta - \theta_\star\|_{\alpha(a,\theta)aa^\mathsf{T}}^2 \quad \text{where} \quad \alpha(a, \theta) = \int_0^1 (1 - t)\dot{\mu}(\langle a, \theta(t) \rangle)dt\,.$$

*Moreover, there exists a real number $\zeta(a, \theta) \in \{\langle a, \theta(t) \rangle \colon t \in [0, 1]\}$ such that*

$$\dot{\mu}(\zeta(a, \theta)) = 2\alpha(a, \theta)\,.$$

*Proof.* By Lemma 30, for any $(y, a) \in [0, 1] \times \mathbf{B}_2^d$,

$$\ell(y, \mu(\langle a, \theta \rangle)) - \ell(y, \mu(\langle a, \theta_\star \rangle)) = (\mu(\langle a, \theta_\star \rangle) - y)\langle a, \theta - \theta_\star \rangle + \alpha(a, \theta)\langle aa^\mathsf{T}(\theta - \theta_\star), \theta - \theta_\star \rangle\,.$$

Integrating both sides with respect to $P_a(dy)$ and noting that, by our realisability assumption, $\int y P_a(dy) = \mu(\langle a, \theta_\star \rangle)$, which leads to the first term dropping out. Thus,

$$\int \ell(y, \mu(\langle a, \theta \rangle)) - \ell(y, \mu(\langle a, \theta_\star \rangle)) P_a(dy) = \|\theta - \theta_\star\|_{\alpha(a,\theta)aa^\mathsf{T}}^2\,.$$

For the second result, repeat the argument with the Lagrange form of the remainder. ∎

### F.2.1 Proof of the upper bound on the eluder dimension bound, Proposition 4

The following proposition is a special case of proposition 8 of Sun and Tran-Dinh (2019). The lemma thereafter is a simple numerical inequality that will come in handy.

**Proposition 34.** *Let $\mu \colon U \to [0, 1]$ be an $M$-self-concordant link function. Then, for any $u, u' \in U^\circ$ satisfying $|u - u'| \le c$, $\dot{\mu}(u) \le \exp(cM)\dot{\mu}(u')$.*

**Lemma 35.** *Suppose that $a > 1$, $x \ge 1$, $b \ge 0$ and $a^x \le bx + 1$. Then,*

$$x \le \log(1 + b/\log(a))/\log(a)\,.$$

*Proof of Lemma 35.* Let $f(x) = a^x$ and $g(x) = bx + 1$. Since $f$ is convex and $g$ is affine, they intersect at no more than two points. Since they intersect at 0, we have that the set of $x$ satisfying $f(x) \le g(x)$ is of the form $[0, y]$ for some $y \ge 0$. Now, let $y' = \log(1 + b/\log(a))/\log(a)$. Then, a quick calculation shows that $f(y') > g(y')$. Thus, $y' > y$. ∎

**Definition 7** (Witnesses and witness sequences). Fix a function class $\mathcal{G} \subseteq \mathbf{R}^{\mathcal{A}}$, a scale $\omega > 0$, and a sequence of actions $(a_1, \dots, a_k) \in \mathcal{A}^k$. A sequence $(g_1, \dots, g_k) \in \mathcal{G}^k$ is called an $\omega$-*witness sequence* for $(a_1, \dots, a_k)$ if for every $t \in \{1, \dots, k\}$,

$$\sum_{i=1}^{t-1} g_t(a_i) \leq \omega \quad \text{and} \quad g_t(a_t) \geq \omega .$$

In this case, $g_t$ is called a *witness* for $a_t$ (given the prefix $a_1, \dots, a_{t-1}$). When $\mathcal{G} = \bar{\Phi}(\mathrm{GLM}(\mu, \Theta))$, we will write witnesses as parameters $(\theta_1, \dots, \theta_k) \subseteq \Theta$ by identifying $g_t(\cdot) = \bar{\varphi}(\cdot, \theta_t)$.

The following is the final lemma used in the proof of Proposition 4.

**Lemma 36.** *Let* $(a_1, \dots, a_k)$ *be an* $\omega$-*independent sequence with respect to* $\bar{\Phi}(\mathrm{GLM}(\mu, \Theta))$, *and let* $\theta_1', \dots, \theta_k'$ *be the corresponding witnesses. Then, there exists a witness sequence* $(\theta_1, \dots, \theta_k) \subset \Theta$ *such that*

$$|\langle a_i, \theta_t - \theta_\star \rangle| \leq 2\sqrt{\kappa \omega} \quad \text{for all} \quad i \leq t \leq k .$$

*In particular, if* $\omega \leq 1/(4\kappa M^2)$, *then*

$$|\langle a_i, \theta_t - \theta_\star \rangle| \leq 1/M \quad \text{for all} \quad i \leq t \leq k .$$

*Proof.* Since $\theta_t'$ is a witness to the eluder condition for $a_t$, we have that

$$\bar{\varphi}(a_t, \theta_t') \geq \omega , \tag{9}$$

$$\sum_{i=1}^{t-1} \bar{\varphi}(a_i, \theta_t') \leq \omega . \tag{10}$$

Observe that the map $\theta \mapsto \bar{\varphi}(a, \theta)$ is convex for all $a \in \mathcal{A}$, and is minimised at $\theta_\star$, where $\bar{\varphi}(a, \theta_\star) = 0$ for all $a \in \mathcal{A}$. Therefore, there exists $\lambda_t \in [0, 1]$ such that

$$\bar{\varphi}(a_t, \theta_\star + \lambda_t(\theta_t' - \theta_\star)) = \omega .$$

Define $\theta_t = \theta_\star + \lambda_t(\theta_t' - \theta_\star)$, which is an element of $\Theta$ since $\Theta$ is convex and contains $\theta_\star$ and $\theta_t'$. We will show that $\theta_t$ is the desired witness. Observe that $\theta_t$ satisfies Eq. (9) with equality. To see that $\theta_t$ also satisfies Eq. (10), we have

$$\sum_{i=1}^{t-1} \bar{\varphi}(a_i, \theta_t) \leq \sum_{i=1}^{t-1} \bar{\varphi}(a_i, \theta_t') \leq \omega ,$$

where the first inequality follows from the convexity of $\theta \mapsto \bar{\varphi}(a_i, \theta)$ and the fact that $\theta_t$ is on the line segment connecting $\theta_\star$ and $\theta_t'$, and the second inequality follows from Eq. (10). Therefore, $\theta_t$ satisfies the eluder condition. Next, we have

$$\bar{\varphi}(a_j, \theta_t) \leq \omega , \qquad \forall j \leq t . \tag{11}$$

Furthermore, by Lemma 33, there exists a real number $\zeta_{j,t}$ on the interval connecting $\langle a_j, \theta_t \rangle$ and $\langle a_j, \theta_\star \rangle$ such that

$$\bar{\varphi}(a_j, \theta_t) = \frac{1}{2} \dot{\mu}(\zeta_{j,t}) \langle a_j, \theta_t - \theta_\star \rangle^2 \geq \frac{1}{2\kappa} \cdot \langle a_j, \theta_t - \theta_\star \rangle^2 ,$$

which together with Eq. (11) implies that

$$|\langle a_j, \theta_t - \theta_\star \rangle| \leq \sqrt{2\kappa \omega} \leq 2\sqrt{\kappa \omega} .$$

$\blacksquare$

**Proposition 4.** *Let Assumption 1 hold. Then, there exists a universal constant $C > 0$ such that*

$$\dim_{\mathrm{elud}}^{\sigma}(\varepsilon; \bar{\Phi}(\mathrm{GLM}(\mu, \Theta))) \leq Cd \log(1 + S^2 L/\varepsilon) \quad \text{for all} \quad 0 < \varepsilon \leq \sigma \leq 1/(4\kappa M^2) .$$

*Proof of Proposition 4.* Let $a_1, \ldots, a_k$ and $\theta'_1, \ldots, \theta'_k$ be witness to the eluder dimension in question, in that they satisfy

$$\sum_{i=1}^{t-1} \bar{\varphi}(a_i, \theta'_t) \leq \omega \quad \text{and} \quad \bar{\varphi}(a_t, \theta'_t) \geq \omega$$

for some $\omega \in [\varepsilon, \sigma]$ and all $t \leq k$. By Lemma 36, there exists an alternative witness sequence $(\theta_1, \ldots, \theta_k) \subset \Theta$ such that for all $t \leq k$ and all $i \leq t$, we have

$$|\langle a_i, \theta_t - \theta_\star \rangle| \leq \frac{1}{M}. \tag{12}$$

Next, for any $\lambda \geq 0$, define the positive semidefinite matrix

$$H_{t-1}(\lambda) = \sum_{i=1}^{t-1} \dot{\mu}(\langle a_i, \theta_\star \rangle) a_i a_i^\mathsf{T} + \lambda I$$

For each $i \leq t \leq k$, use Lemma 33 to construct a real number $\zeta_{i,t}$ on the interval connecting $\langle a_i, \theta_t \rangle$ and $\langle a_i, \theta_\star \rangle$ that satisfies $\dot{\mu}(\zeta_{i,t}) = 2\alpha(a_i, \theta_t)$. Now, by Eq. (12) for all $i \leq t \leq k$,

$$|\zeta_{i,t} - \langle a_i, \theta_\star \rangle| \leq |\langle a_i, \theta_t - \theta_\star \rangle| \leq \frac{1}{M},$$

we have by Proposition 34 that, for all $i \leq t \leq k$,

$$e^{-1}\dot{\mu}(\langle a_i, \theta_\star \rangle) \leq \dot{\mu}(\zeta_{i,t}) \leq e \cdot \dot{\mu}(\langle a_i, \theta_\star \rangle). \tag{13}$$

Hence, using Lemma 33 and Eq. (13), we have the bound

$$\omega \geq \sum_{i=1}^{t-1} \bar{\varphi}(a_i, \theta_t) \geq \frac{1}{2e} \|\theta_t - \theta_\star\|^2_{H_{t-1}(0)}.$$

Taking $\lambda = \omega/(2S^2)$, this gives

$$\begin{aligned}
\|\theta_t - \theta_\star\|^2_{H_{t-1}(\lambda)} &\leq \|\theta_t - \theta_\star\|^2_{H_{t-1}(0)} + \lambda\|\theta_t - \theta_\star\|^2 \\
&\leq 2e\omega + 4\lambda S^2 \\
&\leq 2\omega(e+1). \tag{14}
\end{aligned}$$

Now, letting $x_t = \dot{\mu}(\langle a_t, \theta_\star \rangle)^{1/2} a_t$, we have that

$$\begin{aligned}
\omega &\leq \bar{\varphi}(a_t, \theta_t) && \text{(definition of } \omega, a_t, \theta_t) \\
&= \frac{\dot{\mu}(\zeta_{t,t})}{2} \langle a_t, \theta_t - \theta_\star \rangle^2 && \text{(definition of } \zeta_{t,t}) \\
&\leq \frac{e}{2} \dot{\mu}(\langle a_t, \theta_\star \rangle) \langle a_t, \theta_t - \theta_\star \rangle^2 && \text{(Eq. (13))} \\
&\leq \frac{e}{2} \dot{\mu}(\langle a_t, \theta_\star \rangle) \|a_t\|^2_{H_{t-1}^{-1}(\lambda)} \|\theta_t - \theta_\star\|^2_{H_{t-1}(\lambda)} && \text{(Cauchy-Schwarz)} \\
&\leq \omega e(e+1) \|x_t\|^2_{H_{t-1}^{-1}(\lambda)}. && \text{(Eq. (14))}
\end{aligned}$$

Whence, we conclude that for all $t \leq k$,

$$\|x_t\|^2_{H_{t-1}^{-1}(\lambda)} \geq (e(e+1))^{-1} =: c.$$

Using this lower bound and the matrix determinant lemma, we have

$$\det H_k(\lambda) = \lambda^d \prod_{t=1}^k (1 + \|x_t\|^2_{H_{t-1}^{-1}(\lambda)}) \geq \lambda^d (1+c)^k.$$

On the other hand, using the AM-GM inequality and that $\|x_t\|^2 = \dot{\mu}(\langle a_t, \theta_\star \rangle)\|a_t\|^2 \leq L$, we have the upper bound

$$\det H_k(\lambda) \leq \left(\frac{\text{tr}(H_k(\lambda))}{d}\right)^d \leq \left(\lambda + \frac{kL}{d}\right)^d.$$

Putting the two inequalities together yields the inequality

$$(1+c)^{\frac{k}{d}} \leq \frac{kL}{d\lambda} + 1.$$

Now, applying Lemma 35 with $a = 1 + c$, $x = k/d$ and $b = L/\lambda = 2S^2 L/\omega$, we obtain

$$k \leq d \log\left(1 + \frac{2S^2 L}{\omega \log(1+c)}\right) / \log(1+c) \leq de^{2e} \log(1 + 2S^2 L e^{2e}/\omega),$$

where the second inequality follows by substituting in the definition of $c$ and using that $e^x(1+e^x) \geq e^{2x}$ for $x \geq 0$. Since the above bound is decreasing with $\omega \geq \varepsilon$, it is maximised at $\omega = \varepsilon$. ∎

### F.3 Regret bound for $\ell$-UCB with the logistic model

**Proposition 5** (Regret for $\ell$-UCB with the logistic model). *Let $\delta \in (0,1)$, $S > 0$ and $n \in \mathbf{N}_+$. Consider the setting of Theorem 1, with the model class $\mathcal{F} = \mathrm{GLM}(\mu, \Theta)$ where $\mu(u) = 1/(1+e^{-u})$ and the logistic loss function $\ell_X$. Consider running $\ell$-UCB with confidence widths $(\beta_t)_{t \in \mathbf{N}_+}$ given by*

$$\beta_t = 5/2 + 60(3S+1)\big[d \log(1+8Sn) + \log(h_t/\delta)\big], \qquad h_t = e + \log(1+t).$$

*Then, for a constant $C > 0$, with probability at least $1 - \delta$, the resulting regret satisfies the bound*

$$R_n \leq C\sqrt{n\eta(a_\star)d\beta_n}\,(1 + \log(1+Sn)) + Cd\beta_n\big[(1 + \log(1+Sn))^2 + e^S S \log(1+S)\big] + 12e^S.$$

*Proof.* By Proposition 15, $c = b + 4$; by Lemma 31, $b = 4S$; by Proposition 32, $N_n \leq (1+8Sn)^d$. Combined with Theorem 1, these results yield the confidence widths

$$\beta_t = \frac{5}{2} + 60(3S+1)\big[d \log(1+8Sn) + \log(h_t/\delta)\big].$$

Recall that for the logistic model, $L = 1/4$, $M = 1$, $\kappa = 3e^S$, and, by Proposition 19, $\gamma = 2/\log_2(e)$. If $n < 4\kappa M^2 = 12e^S$, then by boundedness of the costs,

$$R_n \leq n < 12e^S.$$

Thus, it remains to consider the case $n \geq 12e^S$. Take $\sigma = 1/(4\kappa M^2) = 1/(12e^S)$. Then $\sigma \geq 1/n$, and by Proposition 4 and the discussion immediately thereafter, for some $C > 0$, the $\frac{1}{n}$-eluder dimension satisfies

$$d_n^\sigma = \dim_{\mathrm{elud}}^\sigma\left(\frac{1}{n}, \bar{\Phi}(\mathcal{F})\right) \leq Cd \log(1+Sn).$$

Hence, for some $C' > 0$,

$$\begin{aligned}
\Gamma_n^\sigma &= \gamma\big[1 + (d_n^\sigma + 1)4S + 4d_n^\sigma \beta_n \log(1+4Sn)\big]\\
&\leq C'd\beta_n\big[1 + \log(1+Sn)\big]^2,
\end{aligned}$$

where we used that $\beta_n \geq 5/2$, $\beta_n \geq S$, and $\log(1+4Sn) \leq C(1 + \log(1+Sn))$ for a universal constant $C$. Moreover, again by Proposition 4, for some $C > 0$,

$$\dim_{\mathrm{elud}}^\sigma\big(\sigma, \bar{\Phi}(\mathcal{F})\big) \leq Cd \log(1+S^2\kappa) = Cd \log(1+3e^S S^2) \leq C'dS \log(1+S),$$

for another constant $C' > 0$. Therefore,

$$\left(\frac{4\beta_n}{\sigma} + 1\right)\dim_{\mathrm{elud}}^\sigma(\sigma; \bar{\Phi}(\mathcal{F})) + 1 \leq Cd\beta_n e^S S \log(1+S) + 1$$

for some $C > 0$. Combining these estimates with the upper bound for regret given by Theorem 1, and simplifying the constants, we obtain the claimed result. ∎

# G   Proof of lower bound on the eluder dimension in GLMs, Theorem 2

This section establishes our lower bound on the eluder dimension for generalised linear models. The construction is based on the technique of Dong et al. (2019).

**Theorem 2** (GLM $\ell_1$-eluder dimension lower bound). *Let $(\mu, \ell)$ satisfy the last four properties of Assumption 1 (link $L$-Lipschitz, $M$-self-concordant, link-derivative lower bound, and link–loss compatibility). Fix $S \geq 4/M$ and assume that $[-S, 0] \subset U$. Write*

$$\tilde{\kappa} = \frac{\dot{\mu}(0)}{2\dot{\mu}(-S/2)} \in (0, \infty), \qquad b = \min\{\lfloor S \rfloor, d-1\}.$$

*Then, there exist $\mathcal{A} \subset \mathbf{B}_2^d$, $\Theta \subset S\mathbf{B}_2^d$, $\theta_\star \in \Theta$, and a bandit instance $\mathcal{P} = \{P_a : a \in \mathcal{A}\}$ such that $(\mathcal{A}, \Theta, \mu, \ell)$ satisfy Assumption 1, $\eta(a) = \mu(\langle a, \theta_\star \rangle)$, and $P_a = \delta_{\eta(a)}$ for all $a \in \mathcal{A}$. Writing $\mathcal{F} = \mathrm{GLM}(\mu, \Theta)$, for every $\varepsilon \leq \dot{\mu}(0)/(2M^2)$ the eluder dimension of the associated expected excess-loss class $\bar{\Phi}(\mathcal{F})$ satisfies*

$$\dim_{\mathrm{elud}}^\infty(\varepsilon; \bar{\Phi}(\mathcal{F})) \geq \frac{d-1}{4b} \exp\left\{\min\left(\frac{b}{16}, \frac{(\log(\tilde{\kappa}))_+^2}{8SM^2 + 4(\log(\tilde{\kappa}))_+}\right)\right\},$$

*for a sequence of actions taking values in $\mathcal{A}$, where $(x)_+ := \max\{x, 0\}$.*

Our proof will use the following lemma, given as Lemma A.1 in Du et al. (2020).

**Lemma 37** (Johnson-Lindenstrauss packing lemma). *For any integer $D \geq 2$ and any parameter $\zeta \in (0, 1)$, there exists a finite set $\Phi \subset \mathbf{S}^{D-1} := \{x \in \mathbf{R}^D : \|x\|_2 = 1\}$ with $|\Phi| \geq \lfloor \exp(D\zeta^2/8) \rfloor$ and*

$$|\langle x, y \rangle| \leq \zeta \qquad \text{for all distinct } x, y \in \Phi.$$

*Proof of Theorem 2.* We will choose $\zeta \in (0, 1)$ below, and set

$$N = \lfloor \exp(b\zeta^2/8) \rfloor.$$

If $N = 1$, let $x_1 \in \mathbf{R}^b$ be any unit vector. If $N \geq 2$, let $x_1, \ldots, x_N \in \mathbf{R}^b$ satisfy $\|x_i\| = 1$ for all $i \leq N$ and $|\langle x_i, x_j \rangle| \leq \zeta$ for $i, j \leq N$ with $i \neq j$; such vectors exist by Lemma 37.

Let $e_1, \ldots, e_d$ denote the basis vectors of $\mathbf{R}^d$. Let $m = \lfloor (d-1)/b \rfloor \geq 1$ be the number of length $b$ blocks that fit into the $d-1$ dimensions spanned by $e_2, \ldots, e_d$. Let $E_i : \mathbf{R}^b \to \mathbf{R}^d$ insert $v \in \mathbf{R}^b$ into the coordinates of the $i$th such block; that is, for $i \in [m]$,

$$E_i(v) = \sum_{\ell=1}^b v_\ell e_{1+(i-1)b+\ell}.$$

Define the optimal parameter vector $\theta_\star = -2^{-1/2} S e_1$ such that $\eta(a) = \mu(\langle a, \theta_\star \rangle)$. We take $\Theta$ to be the convex hull of $\theta_\star$ and the vectors

$$\theta_{ij} = \theta_\star + 2^{-1/2} S E_i(x_j), \qquad (i, j) \in [m] \times [N].$$

We take the arm-set $\mathcal{A}$ to consist of the vectors

$$a_{ij} = -\frac{\theta_\star}{S} + 2^{-1/2} E_i(x_j), \qquad (i, j) \in [m] \times [N].$$

With this construction, we have the following properties:

$$
\begin{aligned}
&\langle a_{ij}, \theta_\star \rangle = -S/2 && \forall (i, j) \in [m] \times [N], \\
&\langle a_{ij}, \theta_{ij} \rangle = 0 && \forall (i, j) \in [m] \times [N], \\
&\langle a_{ij}, \theta_{i'j'} \rangle = -S/2 && \forall i, i' \in [m] \text{ with } i \neq i' \text{ and all } j, j' \in [N], \\
&\langle a_{ij}, \theta_{ij'} \rangle \in [-(S/2)(1+\zeta), -(S/2)(1-\zeta)] && \forall (i, j) \in [m] \times [N] \text{ and all } j' \in [N] \setminus \{j\}.
\end{aligned}
$$

In particular, $\mathcal{A} \subset \mathbf{B}_2^d$, $\Theta \subset S\mathbf{B}_2^d$, and every inner product of an action in $\mathcal{A}$ with a vertex of $\Theta$ lies in $[-S, 0] \subset U$; since $\Theta$ is convex, the same holds for all $\theta \in \Theta$. Thus $(\mathcal{A}, \Theta, \mu, \ell)$ satisfy Assumption 1.

Let $P_a = \delta_{\eta(a)}$ for each $a \in \mathcal{A}$, and let $\mathcal{F} = \mathrm{GLM}(\mu, \Theta)$. For each $(i, j) \in [m] \times [N]$, let $\bar{\varphi}_{ij} \in \bar{\Phi}(\mathcal{F})$ denote the expected excess loss comparing $\theta_{ij}$ against $\theta_\star$:

$$\bar{\varphi}_{ij}(a) = \ell(\mu(\langle a, \theta_\star \rangle), \mu(\langle a, \theta_{ij} \rangle)) - \ell(\mu(\langle a, \theta_\star \rangle), \mu(\langle a, \theta_\star \rangle)).$$

Consider the sequence of actions $(a_{ij})$ given by

$$a_{1,1}, \ldots, a_{1,N}, a_{2,1}, \ldots, a_{2,N}, \ldots, a_{m,1}, \ldots, a_{m,N}, \tag{15}$$

where we index by enumerating $[m] \times [N]$ in lexicographic order. We will show that for a suitable choice of $\zeta \in (0, 1)$, (15) is an $\omega$-eluder sequence for

$$\omega = \frac{\dot{\mu}(0)}{2M^2},$$

witnessed by $\{\bar{\varphi}_{ij} \colon (i, j) \in [m] \times [N]\}$. This implies that $\dim_{\mathrm{elud}}^\infty(\omega; \bar{\Phi}(\mathcal{F}))$ is at least the length of (15); since $\dim_{\mathrm{elud}}^\infty(\varepsilon; \bar{\Phi}(\mathcal{F}))$ is non-increasing in $\varepsilon$, the same lower bound then holds for every $\varepsilon \leq \omega$.

*Large deviation at new action.* Let $f(u) = \ell(\mu(-S/2), \mu(u))$. Observe that

$$\dot{f}(-S/2) = \mu(-S/2) - \mu(-S/2) = 0, \qquad \ddot{f}(-S/2) = \dot{\mu}(-S/2),$$

by the link and loss compatibility properties in Assumption 1. By Taylor's expansion with integral remainder, around $-S/2$, and using that $\dot{f}(-S/2) = 0$, we have

$$\bar{\varphi}_{ij}(a_{ij}) = f(0) - f(-S/2) = \int_0^1 (1-t)(S/2)^2 \ddot{f}\left(-\tfrac{S}{2} + \tfrac{tS}{2}\right) dt$$

$$= (S/2)^2 \int_0^1 (1-t)\dot{\mu}\left(-\tfrac{S}{2} + \tfrac{tS}{2}\right) dt.$$

By Proposition 34, we have the bound

$$\dot{\mu}\left(-\tfrac{S}{2} + \tfrac{tS}{2}\right) \geq \dot{\mu}(0) \exp\left(-\tfrac{MS}{2}(1-t)\right).$$

Combined with the previous display, writing $\alpha = \frac{MS}{2}$, this gives

$$\bar{\varphi}_{ij}(a_{ij}) \geq (S/2)^2 \dot{\mu}(0) \int_0^1 (1-t) \exp\left(-\alpha(1-t)\right) dt = \frac{\dot{\mu}(0)}{M^2} \left(1 - \exp(-\alpha)(1+\alpha)\right) \geq \frac{\dot{\mu}(0)}{2M^2} = \omega,$$

where the final inequality uses that $\alpha \geq 2$ (since we assumed $S \geq 4/M$), and that $x \mapsto 1 - e^{-x}(1+x)$ is increasing on $(0, \infty)$.

*Small cumulative deviation.* At index $(i, j)$, the cumulative deviation is

$$\sum_{t=1}^{i-1} \sum_{\ell=1}^N \bar{\varphi}_{ij}(a_{t\ell}) + \sum_{\ell=1}^{j-1} \bar{\varphi}_{ij}(a_{i\ell}) = \sum_{\ell=1}^{j-1} \bar{\varphi}_{ij}(a_{i\ell}),$$

since for every $i' \neq i$ and every $\ell \in [N]$ we have

$$\langle a_{i'\ell}, \theta_{ij} \rangle = \langle a_{i'\ell}, \theta_\star \rangle = -S/2,$$

and hence $\bar{\varphi}_{ij}(a_{i'\ell}) = 0$.

If $(\log(\tilde{\kappa}))_+ = 0$, choose $\zeta \in (0, 1)$ small enough that $N = \lfloor \exp(b\zeta^2/8) \rfloor = 1$. Then the sum above is empty for every $(i, j)$, so (15) is an $\omega$-eluder sequence. Therefore,

$$\dim_{\mathrm{elud}}^\infty(\varepsilon; \bar{\Phi}(\mathcal{F})) \geq m \geq \frac{d-1}{2b} \geq \frac{d-1}{4b}.$$

Since $(\log(\tilde{\kappa}))_+ = 0$, the exponential factor in the statement is equal to 1, and the result follows.

Assume now that $(\log(\tilde{\kappa}))_+ > 0$. By the one-dimensional self-concordance bound applied to the function $u \mapsto \ell(\mu(-S/2), \mu(u))$, for any $\ell < j$,

$$\bar{\varphi}_{ij}(a_{i\ell}) \leq \frac{\dot{\mu}(-S/2)}{M^2} \exp\{MS\zeta/2\},$$

and there are at most $N \leq \exp\{b\zeta^2/8\}$ such terms in the sum. Therefore,

$$\sum_{\ell=1}^{j-1} \bar{\varphi}_{ij}(a_{i\ell}) \leq \frac{\dot{\mu}(-S/2)}{M^2} \exp\{b\zeta^2/8 + M\zeta S/2\}.$$

Since

$$\tilde{\kappa} = \frac{\dot{\mu}(0)}{2\dot{\mu}(-S/2)},$$

for the above sum to be upper bounded by $\omega = \dot{\mu}(0)/(2M^2)$, it suffices that

$$b\zeta^2/8 + M\zeta S/2 \leq (\log(\tilde{\kappa}))_+.$$

Using $b \leq S$, it suffices that

$$S\zeta^2/8 + M\zeta S/2 \leq (\log(\tilde{\kappa}))_+,$$

or equivalently that

$$\zeta^2 + 4M\zeta - \frac{8(\log(\tilde{\kappa}))_+}{S} \leq 0.$$

Thus, it is enough to choose $\zeta$ in the interval

$$0 \leq \zeta \leq 2\left(\sqrt{M^2 + (2/S)(\log(\tilde{\kappa}))_+} - M\right).$$

With $\zeta$ chosen to be the largest feasible value in $(0,1)$, the sequence (15) is an $\omega$-eluder sequence and has length $mN$. Using $\lfloor x \rfloor \geq x/2$ for $x \geq 1$, we get

$$N \geq \frac{1}{2}\exp\{b\zeta^2/8\} \geq \frac{1}{2}\min\left\{\exp\{b/8\}, \exp\left\{\frac{((\log(\tilde{\kappa}))_+)^2}{8SM^2 + 4(\log(\tilde{\kappa}))_+}\right\}\right\}.$$

Since also $m \geq (d-1)/(2b)$, we conclude that

$$\dim_{\text{elud}}^{\infty}(\varepsilon; \bar{\Phi}(\mathcal{F})) \geq mN \geq \frac{d-1}{4b}\exp\left\{\min\left(\frac{b}{16}, \frac{((\log(\tilde{\kappa}))_+)^2}{8SM^2 + 4(\log(\tilde{\kappa}))_+}\right)\right\}.$$

This completes the proof. ∎

