# OpenReview forum: "Eluder dimension: localise it!"
_NeurIPS.cc/2025/Conference — NeurIPS 2025 spotlight_

### Official Review · Reviewer_tRDf · 2025-06-21

**Clarity:** 3
**Significance:** 3
**Originality:** 3
**Rating:** 4
**Confidence:** 2

**Summary:**

The paper proposed a localized $\ell_1$-eluder dimension that can handle the limitation of the standard eluder dimension in the regret bound of the decision-making problem. The authors first give a lower bound of a small-cost bound by using global eluder dimension in generalised bandits to show that the drawback of global eluder dimension can not match the best known upper bound. Based on this motivation,the authors provided UCB-based algorithms for both Bernoulli bandits and online RL problems and established the adaptive, first-order regret bounds based on their localization approach.

**Questions:**

1. The set of functions $\mathcal{F}$ is bounded and positive. Can it extend to negative or unbounded?
2. Could you give some remarks on the last term in the result of Theorem 4.1, $\mathrm{card}${$ t \leq n : f_t \notin \mathcal{F}' $}? Is it normal in previous related work?
3. In line 172, the sentence seems incomplete. $\mathcal{F}^{\prime}(r)=$ {$\mu(\langle\cdot, \theta\rangle): \theta \in \Theta^{\prime}(r)$}  for what?
4. The lower bound in Theorem 4.2 is for the Eluder dimension. Is it possible to provide the lower bound for the regret bound? Or any evidence to show how good your regret upper bound is?

**Ethical Concerns:**

["NO or VERY MINOR ethics concerns only"]

**Final Justification:**

I appreciate the author's response to improve my understanding of tightness. And my concerns are solved by the authors' rebuttal. The paper provided solid and interesting theoretical contributions, as other reviewers also mentioned. But it lacks experimental results, thus I would like to keep my score.

**Limitations:**

yes

**Quality:**

3

**Strengths And Weaknesses:**

Strengths:
1. The paper provided solid theoretical contributions and fundamental insights. It is good to first provide Theorem 4.2 as evidence to show the disadvantages of the previous method and then introduce localised eluder dimension to show how localization helps it, which is also conceptually meaningful.
2. The writing is good. And it is good to give some function examples to Assumption 2.1. And the authors give some sufficient comparisons with previous related work, such as section 4.1.
3. The paper studied both bandits and RL problems under this method, which shows the efficiency of the method and its potential to be used in more problems, such as RLHF.

Weaknesses:
The paper lacks experimental results. And it is unclear how to use the localised eluder dimension in practice. I understand the main contribution is on the theoretical side, but some empirical results would show the practical benefits of localisation.

---

> ### Author Rebuttal · Authors · 2025-07-30
>
> We thank the reviewer for their careful read and helpful suggestions.
>
> **Localized eluder dimension.** The localised eluder dimension is a purely analytic tool used to prove tighter regret bounds, and is not needed to run the $\ell$-UCB algorithm. Thus, there is no need to compute it.
>
> **The function class $\mathcal{F}$.** The set of functions F do not need to be bounded or positive. The losses however need to be bounded, with known bounds. Hell will freeze over before the requirement that the losses are bounded from below (with a known bound) is removed: this is a feature of every small cost bound in the literature. And by the time this lower bound is known, we might as well offset by the lower bound and say that the loss is nonnegative. The assumption of an upper bound is not necessary: it can be replaced by, for example, the assumption that the right (upper) tail of the losses satisfies a moment condition (e.g. is subgaussian) for any choice of states/actions.
>
> **The number of rogue steps and tightness.** The number of rogue steps (the card{} term) appears only additively and therefore is second order term— negligible for sufficiently large n (the horizon). Similar additive terms also appear in previous works on logistic bandits (which is an special case of our analysis), making this behavior consistent with existing literature (see, e.g., [1]). When specialized to the generalized linear bandit setting our regret bound matches the best bounds for the problem-specific algorithms, and indeed matches the lower bound of [1].
>
> **Experimental scope.** Since the main focus of our work is the theoretical analysis of optimistic algorithms (i.e., eluder dimension) rather than proposing new algorithms, we focus on a comprehensive theoretical treatment of adaptive algorithms for decision making. However, given the reviewers’ interest, we will consider adding a synthetic experimental section in the final version.
>
> [1] Abeille, Marc, et al. "Instance-wise minimax-optimal algorithms for logistic bandits."

---

> > ### Comment · Reviewer_tRDf · 2025-08-05
> >
> > Thanks for your answer. My concerns are solved.

---

### Official Review · Reviewer_9PY3 · 2025-06-25

**Clarity:** 3
**Significance:** 3
**Originality:** 3
**Rating:** 5
**Confidence:** 4

**Summary:**

This paper investigates the limitations of the standard eluder dimension in achieving first-order regret bounds for GLMs in online decision-making. The authors show that exisiting eluder-based approaches inherently suffer from dependence on a global worst-case complexity term \kappa, which cancels out potential gains in small-cost settings. To address this, they propose a localized \ell_1-eluder dimension that captures the complexity near the optimal predictor. They use this refined measure to derive genuinely first-order, instance-adaptive regret bounds in both stochastic bandits and finite-horizon RL, instantiated via new algorithms (\ell-UCB and \ell-GOLF). Theoretical results show improved guarantees over prior work without requiring warm-up phases or likelihood assumptions.

**Questions:**

How restrictive is triangle condition in practice? Are there other loss functions beyond log and Poisson loss that satisfy it?

Could your framework be extended to misspecified setting?

Do you envision the localized loss-based analysis extending to more general function classes, like RKHS or NNs?

**Ethical Concerns:**

["NO or VERY MINOR ethics concerns only"]

**Final Justification:**

The rebuttal explains the necessity of the assumptions in their main theorem and admits to add synthetic experiments, so I deicide to keep the same score.

**Limitations:**

Yes.

**Quality:**

3

**Strengths And Weaknesses:**

Strengths:

The paper identifies and rigorously proves a key limitation in prior work using global eluder dimensions and proposes a novel and solid fix.

The paper achieves the first true first-order bounds in episodic MDP under bounded cost, improving over existing works which suffer from \kappa-dependent regret.

The work applies to both stochastic bandit and online RL settings,with insights that may extend to other model classes and loss structures.

Weaknesses:

The paper is entirely theoretical. While this is acceptable given the depth of the contribution, even a small synthetic experiment would help illustrate the practical gains from localization.

The regret bounds rely on specific assumptions about the loss function (e.g. triangle condition), realizability and self-concordance. These may not hold in broader settings thus it would be helpful to know how sensitive the results are to violations of these assumptions.

Optimization over localized confidence sets can be intractable in general, and the paper only briefly references convex relaxations without discussing implementation feasibility.

---

> ### Author Rebuttal · Authors · 2025-07-30
>
> We appreciate the reviewer’s time and thoughtful feedback on our work.
>
> **Broader Settings.** Perhaps the situation would be more clear if we go through the role of each assumptions.
> 1. Self-concordance: This is a property of the loss function. Since the loss function is chosen by the algorithm designer, one can select a loss that is self-concordant. Many commonly used losses (e.g., logistic and Poisson and many more [1]) satisfy this property. Importantly, self-concordance is not required for our main theoretical result—it is rather used to compare with existing results in specilized settings such as logistic bandits, and to achieve computational efficiency through ellipsoidal relaxtion of the confidence set.
> 2. Triangle condition: This condition seems essential for achieving small-cost bounds. For example, the squared loss does not satisfy the triangle condition, and we know that small-cost regret bounds are not achievable with it (see [2]).
> 3. Realizability: This is a true modeling assumption. Removing it entirely is challenging; however, one could restate our results in terms of a “best-in-class” comparator and derive the corresponding regret bounds. The analysis of such misspecified settings would be possible but is beyond the scope of this paper.
>
> **Computational efficiency.** Appendix C.5 describes an ellipsoidal relaxation of our confidence set (with a $2(1+SM)$ blowup), which reduces the computational complexity to that of LinUCB—a standard approach in the bandit literature. In practice, LinUCB requires solving a joint optimization over the parameter $\theta$ and action $a$, where $\theta$ lies in an ellipse. This optimization is computationally efficient for many common action sets, including finite sets, polytopes, and various other convex sets.
>
> **Experimental scope.** Since the main focus of our work is the theoretical analysis of optimistic algorithms (i.e., eluder dimension) rather than proposing new algorithms, we focus on a comprehensive theorectial treatment of adaptive algorithms for decision making. However, given the reviewers’ interest, we will consider adding a synthetic experimental section in the final version.
>
> **General function classes.** Yes, our analysis indeed naturally extend to general function classes. One should only provide the necessary "supervised learning" fast rate guarantees (See, e.g., [3] for the RKHS case).
>
> [1] Liu, Shuai, et al. "Almost Free: Self-concordance in Natural Exponential Families and an Application to Bandits."
>
> [2] D. J. Foster and A. Krishnamurthy. "Efficient first-order contextual bandits: prediction, allocation, and triangular discrimination."
>
> [3] Akhavan, Arya, et al. "Bernstein-type dimension-free concentration for self-normalised martingales."

---

> > ### Comment · Reviewer_9PY3 · 2025-08-07
> >
> > Thanks for the rebuttal, it addresses my concerns and I keep the same score.

---

### Official Review · Reviewer_5GDg · 2025-07-03

**Clarity:** 2
**Significance:** 3
**Originality:** 3
**Rating:** 5
**Confidence:** 3

**Summary:**

The argument in the paper (both the bandit and RL case) proceeds as:

1. There is a pathological information constant $\kappa$ that nullifies the benefit of small cost regret bounds in prior work.

2. But actually, we should be able to do with a much smaller constant so the above is not a "real effect," but an artifact of the worst-case nature of the previous analysis.

3. To make this tighter, the authors replace with the local constant $\kappa$ but then need to pay an additional term for mis-approximating the actual \kappa constant.. and we are guaranteed that this is generally bounded for good algorithms that eventually hang out in a ball around the true \eta.

**Questions:**

See weaknesses above

**Ethical Concerns:**

["NO or VERY MINOR ethics concerns only"]

**Final Justification:**

My concerns were mostly addressed by the rebuttal. However, I think it is important that the authors provide further intuition about when the constant $\kappa$ blows up, so that the paper is more interpretable and intuitively appreciable.

**Limitations:**

Yes

**Quality:**

3

**Strengths And Weaknesses:**

I thank the authors for the interesting paper and I enjoyed reading it.


### Strengths

**Significance and Originality** The authors identify an important problem in the prior theoretical RL literature and come up with a nice way to tighten the bound and make it genuinely instance-dependent.

**Quality** The mathematical presentation is technically sound.


### Weaknesses

**Literature** The authors overall do a good job of situating the paper. However, they do not comprehensively cover the related area of instance-dependent guarantees in RL that use experiment design-based approaches. Some relevant papers to compare with:

https://arxiv.org/abs/2207.02575
https://arxiv.org/abs/2406.06856
https://arxiv.org/abs/2311.05638

I would be curious to understand how the bound the authors obtain relate to the upper and lower bounds in this line of work, since it seems that the experiment-design style of analysis does not suffer from the limitations that the authors aim to address.

**Writing** While the technical results are nice, some improvements in the writing could definitely help the central insights shine through more; currently, the insights are hidden inside a lot of exposition. For instance, a couple things that would be good to do:

- Have a concrete and easy-to-understand motivating example in the beginning which shows why the information gain factor easily blows up. And make more explicit what sorts of instances this effect is predominant in.

- Highlight the additional technical challenges for the RL case as compared to the bandit case. And highlight this result more as the strongest result of the paper. It could be good to state this inside a theorem environment even if the statement is informal.

---

> ### Author Rebuttal · Authors · 2025-07-29
>
> We appreciate the reviewer’s reading of our work and their helpful suggestions.
>
> **Experimental design-based approaches.** We thank the reviewer for highlighting these relevant papers on experimental design–based approaches. We will include them in the related works section of the final version.
>
> **Comparison with similar results.** Perhaps the explanation in Section 4.1 helps clarify this point. Our analysis removes the need for an explicit warm-up phase, which can be thought of as a forced experimental design step. Instead, we introduce the localized eluder dimension and handle the "experimental design" aspect analytically rather than algorithmically. Thus, our method achieves comparably strong instance-dependent regret bounds without requiring a warm-up, naturally generalizing beyond bandit settings to reinforcement learning scenarios where explicit warm-ups or optimal-design methods are less feasible.
>
> **Writing and examples.** We appreciate the feedback and will work to better highlight the importance of the results in the final version. Regarding the motivating example, we would like to mention that the current example (“clickthrough rates in online advertising,” lines 164–169) is intended to address this point and provide an intuitive illustration. Nevertheless, we agree it would be helpful to further clarify why the information-gain factor can easily blow up, and we will make these scenarios more explicit in the final version.
>
> **Reinforcement learning.** The RL result is indeed the most interesting; we will use the extra space afforded by the camera ready to include it in the main paper. We will also revisit the story-telling to emphasise the RL result more. Thank you for this feedback.

---

> > ### Comment · Reviewer_5GDg · 2025-08-06
> > **Response to Rebuttal**
> >
> > I thank the authors for their response and I will maintain my positive recommendation

---

### Official Review · Reviewer_1B4q · 2025-07-05

**Clarity:** 3
**Significance:** 2
**Originality:** 2
**Rating:** 4
**Confidence:** 3

**Summary:**

This paper aims at overcoming a key limitation in standard regret analysis in standard regret analysis for generalized linear models, where traditional methods fail to achieve first-order regret bounds due to worst-case dependencies. The authors incorporate a new concept called the localized ℓ₁-eluder dimension, enabling sharper, instance-dependent regret bounds. The proposed method improves upon classical results for Bernoulli bandits and extends to reinforcement learning, delivering the first true first-order regret bounds in finite-horizon RL. The findings highlight how localization techniques can unlock more adaptive algorithms, paving the way for further advances in learning theory.

**Questions:**

1.	While the localized eluder dimension eliminates k-dependence, Proposition 4.4 shows rogue steps still depend on the parameter k. Could k still significantly impact performance in extreme cases (e.g., very large k)?
2.	How should key hyperparameters like localization radius r be chosen? Are there adaptive methods to set r?
3.	While 2D bandit experiments demonstrate advantages, would similar improvements hold for higher-dimensional RL tasks?
4.	Have you considered comparisons with non-eluder dimension methods?
5.	Could the authors provide some experimental results on RL benchmarks? Should the authors make the computational complexity of the proposed method?

**Ethical Concerns:**

["NO or VERY MINOR ethics concerns only"]

**Limitations:**

Here the two suggestions for the authors: (i) The theoretical analysis currently focuses on bounded-cost scenarios. The authors should discuss whether/how the method could generalize to unbounded reward settings. (ii) Computational complexity of the localized eluder dimension calculation isn't analyzed. Some discussion of practical scalability would strengthen the work.

**Paper Formatting Concerns:**

The paper is well-written, standard formatting conventions.

**Quality:**

3

**Strengths And Weaknesses:**

Strengths:
1.	The paper makes significant theoretical contributions, with rigorous proofs supporting the proposed method in manuscript and appendix.
2.	The approach is well-suited for online RL tasks, eliminating the need for impractical exploration mechanisms required by prior methods.

Weaknesses:
1.	While the theoretical analysis is thorough, the paper lacks sufficient experimental validation (e.g., RL benchmarks or real-world datasets).
2.	The computational efficiency of the method, especially in high-dimensional state-action spaces, remains unexplored.
3.	The writing could be improved for clarity and style. For instance, the title uses inconsistent capitalization (e.g., "Localise It!")

---

> ### Author Rebuttal · Authors · 2025-07-29
>
> We thank the reviewer for their reading and helpful suggestions.
>
> **Number of rogue steps.** The number of rogue steps appears only additively in the regret bound of Theorem 4.1 and is therefore a second-order term. Hence, even when $\kappa$ is large, its effect remains negligible for sufficiently long horizons.
>
> **Choice of localization radius r.** The radius r is used solely in the analysis and does not need to be specified for the algorithm. For example, setting r=1/M, where M is the self-concordance parameter, ensures that the localized eluder dimension is bounded, as as well as the number of rogue steps, as stated in Proposition 4.3 and Proposition 4.4.
>
> **Experimental scope.** Since the main focus of our work is the theoretical analysis of optimistic algorithms (i.e., eluder dimension) rather than proposing new algorithms, we focus on a comprehensive theoretical treatment of adaptive algorithms for decision making. However, given the reviewers’ interest, we will consider adding a synthetic experimental section in the final version.
>
> **Non eluder methods.** Small-cost regret bounds exist in specific settings (e.g., logistic bandits) without using the eluder dimension. However, the eluder dimension remains a popular complexity measure in bandits and RL and as we show, when applied naively it prevents achieving small-cost bounds. Our contribution introduces localization to overcome this limitation, bringing eluder-based analysis closer to the behavior seen in specialized non-eluder approaches.
>
> **Computational efficiency.** Appendix C.5 describes an ellipsoidal relaxation of our confidence set (with a $2(1+SM)$ blowup), which reduces the computational complexity to that of LinUCB—a standard approach in the bandit literature. In practice, LinUCB requires solving a joint optimization over the parameter $\theta$ and action $a$, where $\theta$ lies in an ellipse. This optimization is computationally efficient for many common action sets, including finite sets, polytopes, and various other convex sets.
>
> **Unbounded costs.** While we do not provide a formal analysis for unbounded costs in this paper, we believe that our techniques could extend to settings with sub-Gamma costs (or other standard tail assumptions). We leave a detailed treatment of this extension as an interesting direction for future work.
>
> **Eluder dimension computation.** The localised eluder dimension is a purely analytic tool used to prove tighter regret bounds, and is not needed to run the $\ell$-UCB algorithm. Thus, there is no need to compute it.

---

> > ### Comment · Reviewer_1B4q · 2025-08-08
> >
> > Thank you for taking the time with the concrete response. After reading through the response and the reviews from other reviewers, I stand by my original evaluation.

---

### Comment · Area_Chair_xoEm · 2025-08-04

The authors have provided responses to your reviews. Please first read and acknowledge it. If you need additional clarification or has additional questions for the authors, please also post them ASAP. Thanks.

---

### Decision · Program_Chairs · 2025-09-17

**Decision:**

Accept (spotlight)

**Comment:**

This submission studies genuinely first-order, instance-adaptive regret for generalized linear models and finite-horizon reinforcement learning. The authors identify why global eluder-dimension analyses are bottlenecked by a worst-case information constant and formalize this limitation with a lower-bound argument. They then introduce a localized eluder dimension that focuses analysis near the optimal predictor and instantiate it with UCB-style algorithms for bandits and an episodic RL counterpart, yielding first-order bounds without a warm-up phase and with an additive rogue-step term that remains second order. The paper also presents an ellipsoidal relaxation that reduces the optimization burden to a LinUCB-like routine and clarifies when the global worst-case constant can become large; reviewers found the theory careful and the extension from bandits to RL a notable strength.

While the problem is important and the contribution is well motivated, the work remains entirely theoretical: there are no RL benchmarks or broader empirical studies to illustrate finite-sample behavior. Some assumptions (triangle condition, realizability, and self-concordance used for efficiency) may limit applicability; the rebuttal helpfully scoped their roles, but a discussion of robustness to mild violations would improve practicality.

Recommendation: Accept. The paper pinpoints a real limitation of global eluder analyses and offers a principled localization remedy that delivers first-order bounds for both bandits and episodic RL.